# Transitions in cell potency during early mouse development are driven by Notch

**Sergio Menchero[1], Isabel Rollan[1], Antonio Lopez-Izquierdo[1], Maria Jose Andreu[1], Julio Sainz de Aja[1†], Minjung Kang[2], Javier Adan[1], Rui Benedito[3], Teresa Rayon[1‡], Anna-Katerina Hadjantonakis[2], Miguel Manzanares[1]\***

[1]Centro Nacional de Investigaciones Cardiovasculares Carlos III (CNIC), Madrid, Spain; [2]Developmental Biology Program, Sloan Kettering Institute, New York, United States; [3]Molecular Genetics of Angiogenesis Group, Centro Nacional de Investigaciones Cardiovasculares Carlos III (CNIC), Madrid, Spain

**Abstract** The Notch signalling pathway plays fundamental roles in diverse developmental processes in metazoans, where it is important in driving cell fate and directing differentiation of various cell types. However, we still have limited knowledge about the role of Notch in early preimplantation stages of mammalian development, or how it interacts with other signalling pathways active at these stages such as Hippo. By using genetic and pharmacological tools in vivo, together with image analysis of single embryos and pluripotent cell culture, we have found that Notch is active from the 4-cell stage. Transcriptomic analysis in single morula identified novel Notch targets, such as early naïve pluripotency markers or transcriptional repressors such as TLE4. Our results reveal a previously undescribed role for Notch in driving transitions during the gradual loss of potency that takes place in the early mouse embryo prior to the first lineage decisions.
DOI: https://doi.org/10.7554/eLife.42930.001

**\*For correspondence:**
mmanzanares@cnic.es

**Present address:** [†]Stem Cell Program, Boston Children's Hospital, Boston, United States; [‡]The Francis Crick Institute, London, United Kingdom

**Competing interests:** The authors declare that no competing interests exist.

## Introduction

The totipotent mammalian zygote has the self-organising capacity of generating embryonic and extraembryonic structures to build a complete organism (*Wennekamp et al., 2013*). This undifferentiated cell will proliferate and its descendants will take lineage decisions that entail a progressive loss of potency. The first differentiation event that leads to distinct lineages takes place during preimplantation development at the morula to blastocyst transition, resulting in the formation of the trophectoderm (TE, extraembryonic population) and the inner cell mass (ICM, that gives raise to the embryonic population and the extraembryonic yolk sac endoderm). How the establishment of these early lineages is achieved has been widely studied and we now know that a combination of morphogenetic cues breaks the symmetry in the embryo (*Cockburn and Rossant, 2010*; *Menchero et al., 2018*; *Sasaki, 2015*). The first morphological sign of differentiation is evident in the compacting morula, 2.5 days after fertilization (embryonic day E2.5), when blastomeres increase their intercellular interactions and outer cells acquire an apical-basal polarity. These polarized cells on the surface enclose an inner group of apolar cells (*Johnson and Ziomek, 1981*; *Ziomek and Johnson, 1980*). The outer versus inner position of the blastomeres correlates with their fate, becoming TE or ICM respectively, although cells can change their position within the embryo (*Anani et al., 2014*; *Tarkowski and Wróblewska, 1967*; *Watanabe et al., 2014*). Prior to compaction, blastomeres appear morphologically equivalent. However, transcriptional differences among blastomeres have been described as early as in the 4 cell embryo (*Burton et al., 2013*; *Goolam et al., 2016*; *Torres-Padilla et al., 2007*). Although cells at this stage are not committed to a specific fate, these early heterogeneities correlate with specific fate biases before lineage commitment. However, how these early heterogeneities arise and their implications in cell plasticity are still unclear (*Chen et al., 2018*).

**eLife digest** We start life as a single cell, which immediately begins to divide to form an embryo that will eventually contain all the different kinds of cells found in the adult body. During the first few rounds of cell division, embryonic cells can become any type of adult cells, but also form the placenta, the organ that sustains the embryo while in the womb. As cells keep on dividing, they lose this ability, called potency, and they take on more specific and inflexible roles.

The first choice embryonic cells must make is whether to become part of the placenta or part of the future body. These types of decisions are controlled by molecular cascades known as signalling pathways, which relay information from the cells surface to its control centre. There, specific genes get turned on or off in response to an outside signal.

Previous research showed that two signalling pathways, Hippo and Notch, help separate placenta cells from those that will form the rest of the body. However, it was not known whether the two pathways worked independently, or if they were overlapping. Menchero et al. therefore wanted to find out when exactly the Notch pathway started to be active, and examine how it helped cells to either become the placenta or part of the future body.

Experiments with developing mouse embryos showed that the Notch pathway was activated after the very first two cell divisions, when the embryo consists of only four cells. Genetic manipulations combined with drug treatments that changed the activity of the Notch pathway confirmed that Notch and Hippo acted independently at this stage. Further, larger-scale analysis of gene activity in these embryos also revealed that Notch signalling was working in a previously unknown way: it turned off the genes that maintain potency, pushing the cells to become more specialised.

Ultimately, identifying this new mode of action for the Notch pathway in the early embryo may help to understand how the signalling cascade acts in other types of processes. This knowledge could be useful, for example, to push embryonic cells grown in the laboratory towards a desired fate.

DOI: https://doi.org/10.7554/eLife.42930.002

Once the embryo compacts, differences in cell membrane contractility and the activity of signalling pathways orchestrate the lineage-commitment of cell populations (*Kono et al., 2014*; *Korotkevich et al., 2017*; *Maître et al., 2016*; *Mihajlović and Bruce, 2016*; *Nishioka et al., 2009*; *Nissen et al., 2017*; *Rayon et al., 2014*). The initial stochastic expression of the main lineage-specific transcription factors (such as CDX2 or GATA3 for the TE, and OCT4 or NANOG for the ICM) is gradually restricted to their definitive domains (*Dietrich and Hiiragi, 2007*; *Posfai et al., 2017*). The Hippo pathway has been shown to act as a readout of cell polarity and therefore, differential intercellular distribution of its components and thus differential activity in polar or apolar cells, will dictate fate (*Cockburn et al., 2013*; *Hirate et al., 2013*; *Leung and Zernicka-Goetz, 2013*; *Wicklow et al., 2014*). In outer cells, the pathway is switched off and the transcriptional coactivator YAP is translocated to the nucleus where it will interact with TEAD4, the effector of the pathway, to promote the expression of key TE genes such as *Cdx2* and *Gata3* (*Nishioka et al., 2009*; *Ralston et al., 2010*). We have previously shown that Notch signalling also has a role in the regulation of *Cdx2*. It is specifically active in the TE, where the intracellular domain of the Notch receptor (NICD) is translocated into the nucleus where it binds to the transcription factor RBPJ to promote target gene expression. Both Notch and Hippo converge on the TEE, an enhancer upstream of *Cdx2* (*Rayon et al., 2014*). YAP/TEAD and NICD/RBPJ transcriptional complexes interact with the chromatin modifier SBNO1 to favour the induction of *Cdx2* (*Watanabe et al., 2017*).

Nevertheless, we still do not understand how these two signalling pathways interact to regulate *Cdx2* in the embryo, if there is crosstalk between them, if they are acting in parallel during development or otherwise. Furthermore, Notch signalling could have other unexplored roles at early stages of mouse development. In this study, we show that Hippo and Notch pathways are largely independent, but that Notch is active earlier, before compaction, and that differences in Notch levels contribute to cell fate acquisition in the blastocyst. Single-embryo RNA-seq points at repressors that block early naïve pluripotency markers as Notch targets. We propose that Notch coordinates the

triggering of initial differentiation events within the embryo and regulates the early specification of the trophectoderm.

## Results

### CDX2 expression in the morula is dependent on the Notch and Hippo signalling pathways

Previously, we have described how Notch and Hippo pathways converge to regulate *Cdx2* expression, and that different allelic combinations for *Rbpj* and *Tead4* lead to a significantly reduced expression of CDX2 (*Rayon et al., 2014*). Notably, we failed to recover double mutant embryos at the blastocyst stage (E3.5), suggesting that the lack of both factors caused lethality before the blastocyst stage. We therefore decided to investigate embryos at the earlier morula stage (E2.5), where we recovered double mutant embryos at Mendelian ratios (*Figure 1—figure supplement 1A*). CDX2 levels were apparently lower in $Rbpj^{-/-};Tead4^{+/-}$ and $Rbpj^{+/-};Tead4^{-/-}$ morulae, as previously observed in blastocysts (*Rayon et al., 2014*). Interestingly, this effect was exacerbated in double mutant embryos ($Rbpj^{-/-};Tead4^{-/-}$) in which we did not detect any CDX2 expression in all embryos of this genotype analysed (*Figure 1A*, *Figure 1—figure supplement 1B*).

Compaction of blastomeres and polarization of outer cells are critical morphological events that take place at the morula stage and are linked to the onset of CDX2 expression (*Ralston and Rossant, 2008*; *Wu et al., 2010*). We therefore decided to investigate if these processes were affected in double mutant morulae. We examined the expression of E-cadherin and phospho-ERM, as markers of cell-cell adhesion and apical polarity. No differences in the distribution or intensity of these markers was observed in any of the allelic combinations examined, including double mutants for *Rbpj* and *Tead4* (*Figure 1—figure supplement 1C*). Therefore, disruption of Notch and Hippo signalling does not alter cellular and morphological events prior to lineage specification, but does result in a dramatic downregulation of *Cdx2* at this stage.

To better understand the contributions of each of the Notch and Hippo pathways to CDX2 expression, we performed correlations in single cells between Notch, YAP and CDX2. We used a transgenic mouse line carrying CBF1-VENUS as a reporter of Notch activity (*Nowotschin et al., 2013*), and we performed immunostaining to detect YAP and CDX2 in morulae and blastocysts from that reporter line. In the blastocyst, the three markers were restricted to nuclei of the TE, while in the morula their expression was more heterogeneous (*Figure 1B*). Nuclear YAP was detected preferentially in outer cells, presumably polarized blastomeres, whereas CBF1-VENUS and CDX2 were detected in both inner and outer cells of the morula. We quantified nuclear intensity levels using a Matlab based segmentation tool, MINS (*Lou et al., 2014*), and found that CBF1-VENUS and YAP both correlated positively with CDX2 at early morula (8–16 cells), late morula (17–32 cells) and blastocyst stages (*Figure 1C*). This correlation was low in the earliest embryos but gradually increased. Interestingly, there was no correlation between CBF1-VENUS and YAP in the early or late morula, suggesting that the two pathways are activated independently at this stage (*Figure 1C*). By the blastocyst stage, these markers did show a positive correlation, albeit weaker than the correlation of either marker with CDX2 (*Figure 1C*). If all three components were taken into account simultaneously, the coefficient of correlation increases in early and late morulae and blastocyst (*Figure 1—figure supplement 2*), indicating that the combination of Notch and Hippo pathways better accounted for CDX2 levels than any of them individually.

In most cases, individual nuclei from morulae were positive for the three markers. However, we did find a few cases in which nuclei were positive for CBF1-VENUS and CDX2 but negative for YAP (*Figure 1B*, arrowhead). We therefore analysed all morulae to determine the distribution of cells positive for each combination of markers. We found that Notch was active in most of the cells at this stage and that the majority of blastomeres were positive for all three of the markers (295 blastomeres, 72.3%; *Figure 1D*). Another noteworthy population was represented by cells that were only positive for CBF1-VENUS and CDX2 (85 blastomeres, 20.8%). However, we rarely found cells expressing YAP and CDX2 but not CBF1-VENUS at the morula stage (*Figure 1D*). Together, this set of experiments shows that Notch and Hippo correlate with CDX2 expression at the morula stage, and suggests that they could be acting independently from each other in its regulation.

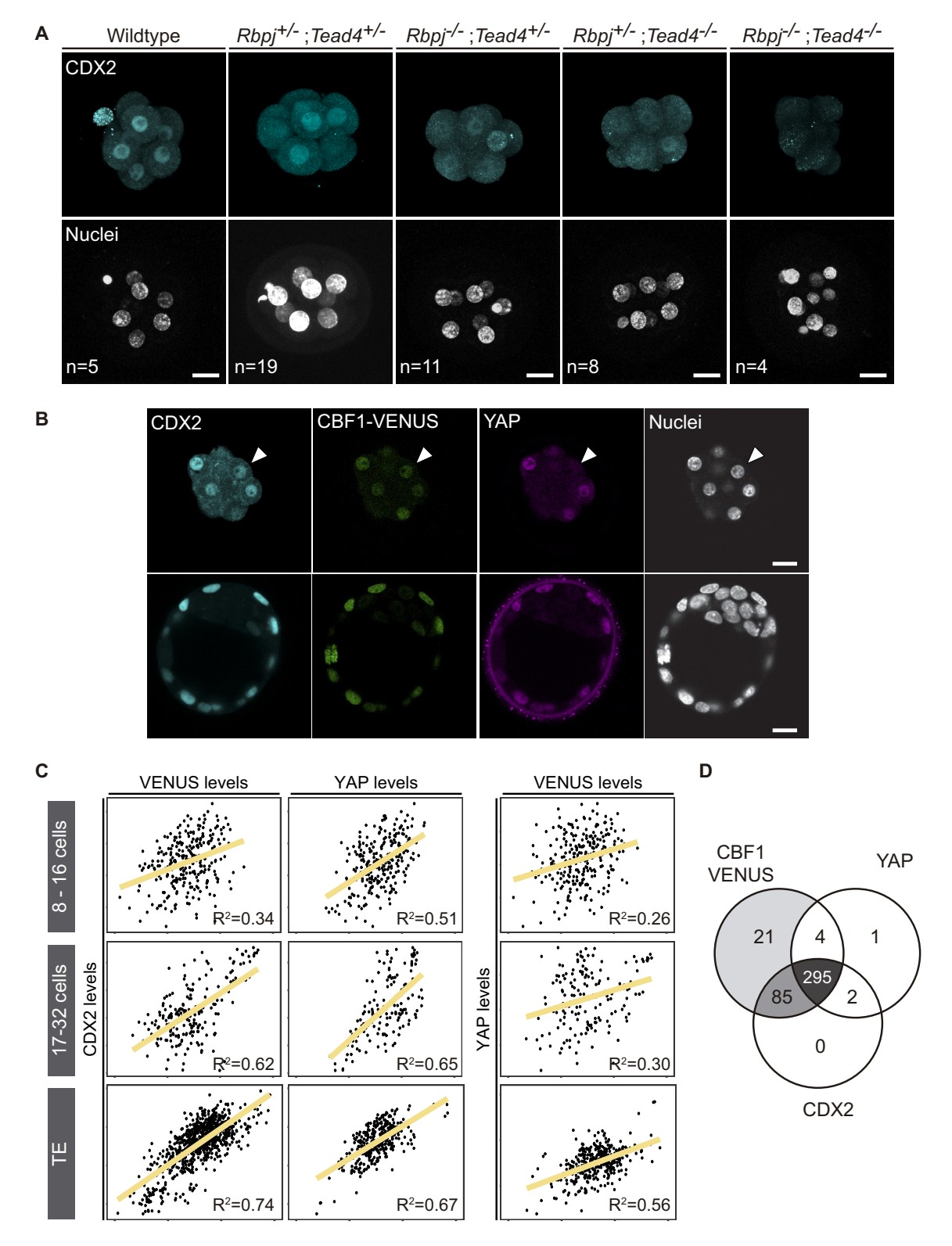

**Figure 1.** CDX2 expression depends on Notch and Hippo inputs. (**A**) Maximal projections of confocal images after immunostaining for CDX2 in different combinations of wildtype and mutant alleles for *Rbpj* and *Tead4* at E2.5. Nuclei were stained with DAPI. Number of embryos (n) is indicated. Scale bars, 20 µm. (**B**) Optical sections of confocal images after immunostaining for CDX2 and YAP in the CBF1-VENUS reporter line at morula (upper row) and blastocyst (lower row) stage. Fluorescent VENUS reporter is directly detected. Arrowheads indicate a cell positive for CDX2 and VENUS, but

*Figure 1 continued on next page*

*Figure 1 continued*

negative for nuclear YAP. Nuclei were stained with DAPI. Scale bars, 20 μm. (**C**) Pairwise correlations of single cell fluorescence intensity levels for CDX2, VENUS and YAP from embryos at early morula (8–16 cells, upper row), late morula (17–32 cells, middle row) and blastocyst (lower row) stage. n = 277 blastomeres from 21 embryos (8–16 cell morulae); n = 211 blastomeres from 12 embryos (17–32 cell morulae); n = 428 blastomeres from six embryos (blastocysts). Person correlation ($R^2$) is indicated for each correlation. (**D**) Venn diagram showing number of positive cells for CBF1-VENUS, YAP and CDX2 at morula stage. n = 415 blastomeres from 28 morulae.

DOI: https://doi.org/10.7554/eLife.42930.003

The following figure supplements are available for figure 1:

**Figure supplement 1.** Lack of *Rbpj* and *Tead4* does not affect compaction or polarization.
DOI: https://doi.org/10.7554/eLife.42930.004
**Figure supplement 2.** Combinatorial input from Notch and Hippo progressively accounts for CDX2 levels from morula to blastocyst.
DOI: https://doi.org/10.7554/eLife.42930.005
**Figure supplement 3.** Hippo components are not affected in Notch loss of function mutant blastocysts.
DOI: https://doi.org/10.7554/eLife.42930.006
**Figure supplement 4.** CBF1-VENUS expression is maintained in *Tead4* mutants.
DOI: https://doi.org/10.7554/eLife.42930.007

## Absence of crosstalk between the Notch and Hippo signalling pathways in the early mouse embryo

The correlation analysis between CBF1-VENUS and YAP expressing blastomeres indicated possible independent roles for Notch and Hippo in the regulation of CDX2 expression (*Figure 1C*). Furthermore, our previous results showed how these two pathways acted in parallel to transcriptionally regulate *Cdx2* through a distal enhancer element (*Rayon et al., 2014*). To further study the interaction between these pathways, we examined TEAD4 and YAP expression in *Rbpj*$^{-/-}$ (*Figure 1—figure supplement 3A,B*) and *Notch1*$^{-/-}$ (*Figure 1—figure supplement 3C,D*) blastocysts. We did not detect any differences in levels or pattern of expression of TEAD4 and YAP either in *Rbpj*$^{-/-}$ or in *Notch1*$^{-/-}$ embryos as compared to wildtype embryos. We also studied the reverse situation, crossing the CBF1-VENUS mouse line as a reporter of Notch pathway activity into the *Tead4* null background. We detected VENUS fluorescent protein in both wildtype and *Tead4*$^{-/-}$ embryos (*Figure 1—figure supplement 4*). Interestingly, CBF1-VENUS expression was maintained in outer cells although the *Tead4*$^{-/-}$ embryos do not form a proper blastocyst (*Nishioka et al., 2008*), in line with previous results showing that some degree of outer identity still is present in *Tead4*$^{-/-}$ embryos (*Yagi et al., 2007*; *Nishioka et al., 2008*; *Frum et al., 2018*). These results confirm that Notch is not required for proper deployment of the transcriptional effectors of the Hippo pathway, and vice versa, that activation of the Notch pathway can occur in embryos deleted for *Tead4*, one of these effectors.

## Notch regulates the onset of *Cdx2* expression

To better understand how parallel signalling pathways drive *Cdx2* expression, we determined if the temporal expression of *Cdx2* was regulated differentially by Notch and Hippo. To do so, we took advantage of pharmacological compounds that allow inhibition of these pathways in a time-controlled manner. We used RO4929097 (RO) to inhibit the Notch pathway (*Münch et al., 2013*) and Verteporfin to block the YAP-TEAD4 interaction (*Liu-Chittenden et al., 2012*). We treated wildtype embryos in two different time-windows: from the two-cell up to morula stage, and from morula to blastocyst. As a control, we treated embryos with DMSO, the solvent used for diluting both inhibitors. This was important, as high doses of DMSO result in embryo lethality, what also limits the concentration of inhibitors to be used. We confirmed that treatment with RO in both time-windows was affecting Notch signalling as expected, as it visibly reduced CBF-VENUS activity in embryos (*Figure 2—figure supplement 1A,B*). Efficacy of Verteporfin in interfering with YAP-TEAD4 activity was inferred by its effect on known targets, as we have previously shown in blastocyst for *Cdx2* (*Rayon et al., 2014*). After treatment, gene expression in embryos was analysed by RT-qPCR. In the early time window, from two-cell to morula, we observed that *Cdx2* was downregulated when Notch was inhibited, while there was no change when Hippo pathway activity was altered (*Figure 2A*). Interestingly, the opposite was found when we modulated the pathways from the morula onwards. *Cdx2* expression was only affected when YAP-TEAD4 activity was blocked (*Figure 2B*). These results show that, although both pathways cooperate in the regulation of *Cdx2*, they act sequentially to

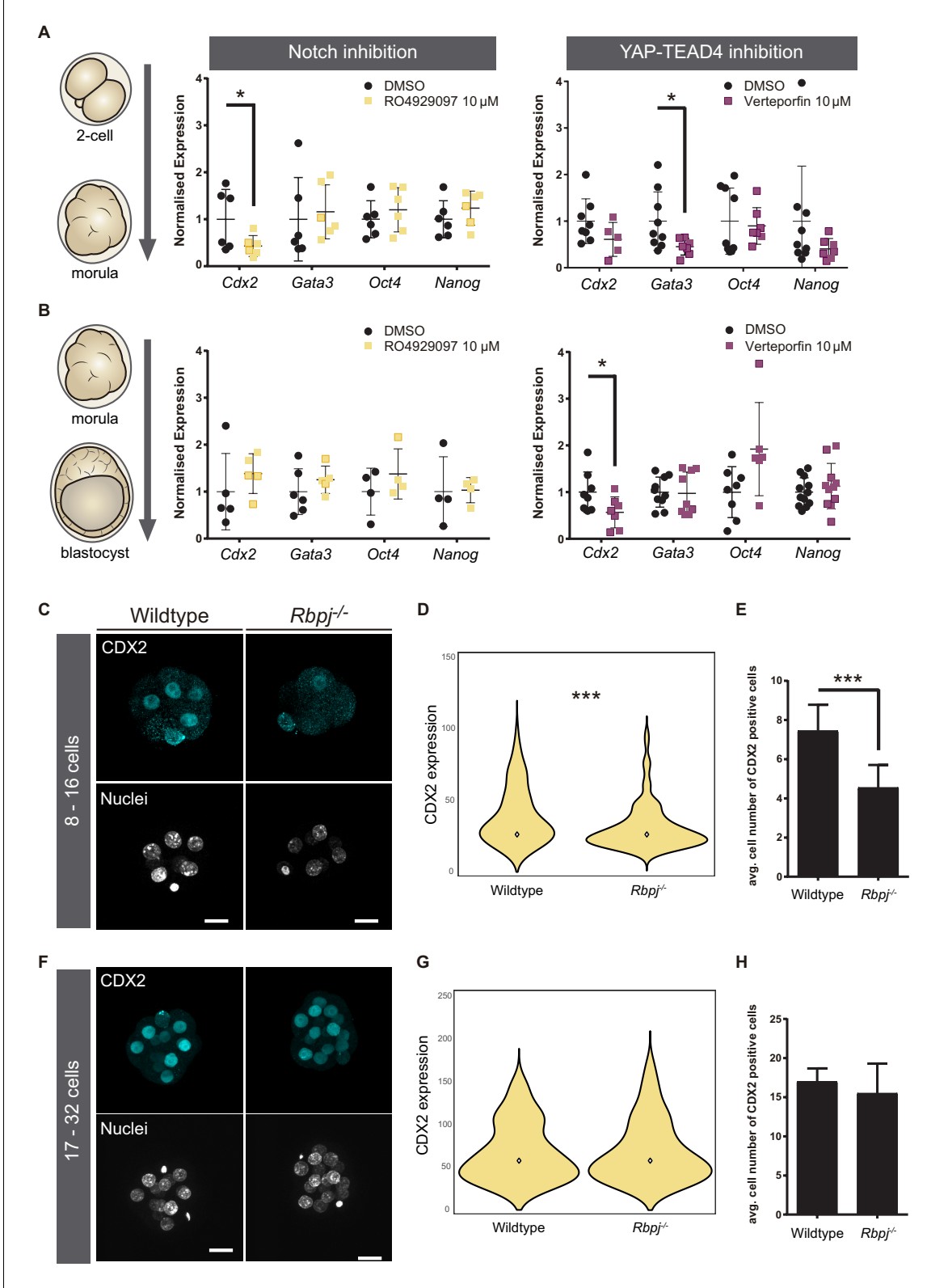

**Figure 2.** Notch regulates CDX2 in the early morula. (A–B) Normalised expression of *Cdx2*, *Gata3*, *Oct4* and *Nanog* in pools of 25 embryos treated with RO4929097 to inhibit Notch (left) or with Verteporfin to inhibit YAP/TEAD interaction (right), from the 2 cell to morula stage (A; Notch inhibition, n = 6); YAP/TEAD inhibition, n = 6–9) or from morula to blastocyst stage (B; Notch inhibition, n = 4–6; YAP/TEAD inhibition, n = 6–11). n represents number of unique pools of 25 embryos. Pools of embryos treated with DMSO were used as controls. * p-value<0.05 by Student's t test. (C) Maximal

*Figure 2 continued on next page*

*Figure 2 continued*

projections of confocal images after immunostaining of CDX2 in wildtype and *Rbpj*$^{-/-}$ early morulae. (**D**) Violin plots of CDX2 intensity levels and (**E**) quantification of number of CDX2 positive cells in wildtype and *Rbpj*$^{-/-}$ early morulae (n = 11 embryos). (**F**) Maximal projections of confocal images after immunostaining of CDX2 in wildtype and *Rbpj*$^{-/-}$ late morulae. (**G**) Violin plots of CDX2 intensity levels and (**H**) quantification of number of CDX2 positive cells in wildtype and *Rbpj*$^{-/-}$ late morulae (wildtype, n = 4 embryos; *Rbpj*$^{-/-}$, n = 5 embryos). Nuclei were stained with DAPI. Scale bar, 20 μm. Data are means ± s.d. ***p<0.001 by Student's t test (**D**) or by Fisher's exact test (**E**).
DOI: https://doi.org/10.7554/eLife.42930.008
The following figure supplements are available for figure 2:
**Figure supplement 1.** CBF1-VENUS levels decrease upon Notch inhibition.
DOI: https://doi.org/10.7554/eLife.42930.009
**Figure supplement 2.** Downregulation of CDX2 in *Notch1*$^{-/-}$ early morulae.
DOI: https://doi.org/10.7554/eLife.42930.010

regulate *Cdx2* levels in a stage specific manner rather than being redundant. Interestingly, *Gata3*, a known target of YAP-TEAD4 independent of *Cdx2* (*Ralston et al., 2010*) is downregulated by Verteporfin in embryos treated from the two-cell to morula stage (*Figure 1A*). This suggests that certainly Hippo signalling acts differently on its various targets, as are *Cdx2* or *Gata3*. *Oct4* and *Nanog* were not significantly changed after Notch or YAP inhibition in any of the time windows.

Next, we wished to confirm these observations in morula stage embryos using genetic loss of function models. We recovered early (8–16 cells) and late (17–32 cells) morulae and analysed CDX2 expression in wildtype and *Rbpj*$^{-/-}$ embryos (*Figure 2C–E*). We found that *Rbpj*$^{-/-}$ early morulae had a significantly lower level of nuclear CDX2 expression (*Figure 2D*) and number of CDX2 positive cells compared to control littermates (*Figure 2E*). In contrast, we did not observe differences at the late morula stage (*Figure 2F–H*). The same observations were obtained when we analysed embryos from another mutant for the pathway, *Notch1*$^{-/-}$: early morulae (8–16 cells) showed lower CDX2 expression and a decrease in the number of CDX2 positive cells (*Figure 2—figure supplement 1A–C*), but late (17–32 cells) morulae did not (*Figure 2—figure supplement 1D–F*). This result is interesting, as it suggests that *Notch1* is the main receptor acting upstream of RBPJ during preimplantation development as its loss is enough to recapitulate *Rbpj* loss of function effects.

These results indicate that there is an earlier requirement for Notch than for Hippo in the regulation of *Cdx2*, and that both pathways exert non-redundant roles. Our observations are suggestive of a model where Notch regulates the onset of *Cdx2* expression, and the Hippo pathway subsequently maintains its expression.

## The Notch pathway is heterogeneously active in the embryo starting at the 4-cell stage

In light of the above findings revealing a requirement of the Notch pathway for the early stages of mouse preimplantation development, we decided to investigate when Notch is first active, using the CBF1-VENUS reporter line as a transcriptional readout of the pathway. We recovered embryos from the CBF1-VENUS line and found that the reporter was first active in 4 cell embryos, albeit at lower levels than at later stages (*Figure 3A*). The number of VENUS positive cells was variable among embryos, with at least a third of embryos examined having no positive cells (7 out of 20; *Figure 3—figure supplement 1*). This strongly suggests that the onset of Notch pathway activation is indeed occurring at this stage. As a general rule, the number of positive blastomeres increased with the total number of cells per embryo (*Figure 3B*; *Figure 3—figure supplement 1*). In the compacted morula, most of the cells were positive, but the activity of the reporter was quickly restricted to the outer TE cells once the blastocyst formed (*Figure 1B*).

In order to follow the dynamics of the reporter and determine how restriction of Notch activity is achieved during development, we performed live imaging for up to 24 hr of embryos from the compacted morula (16 cell) to the early blastocyst stage (*Video 1*, *Figure 3C*). After tracking of the cells in each embryo (n = 7; *Figure 3—figure supplement 2A*), we used a Matlab based tool to analyse the behaviour of each individual cell and its progeny within the embryo. With this tool, we were able to reconstruct the embryo in each time point and assign an initial position (inner or outer) to each blastomere as well as its final location in the TE/out or the ICM/in (*Video 2*, *Figure 3D*). We first generated a lineage tree so that each lineage or family includes a cell in the time frame 0 and all

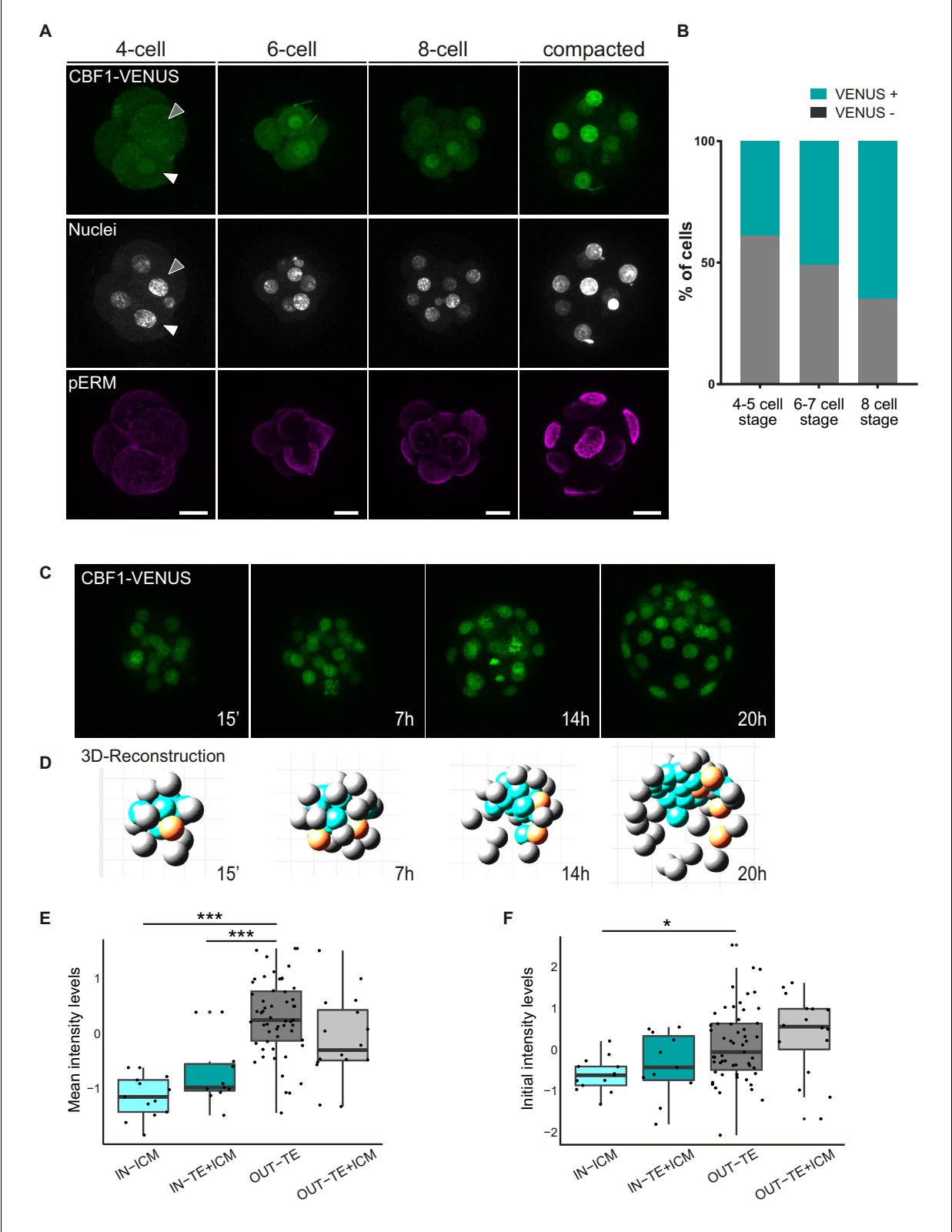

**Figure 3.** CBF1-VENUS dynamics in the mouse preimplantation embryo. (**A**) Maximal projections of confocal images of CBF1-VENUS reporter line in 4 cell, 6 cell, 8 cell and compacted morula stages. A white and a grey arrowhead respectively indicate a positive and a negative cell for VENUS in a 4 cell embryo. Immunostaining of pERM (bottom row) confirms acquisition of apical polarity in compacted morulae. Nuclei were stained with DAPI. Scale bar, 20 μm. (**B**) Percentage of VENUS positive cells per embryo at different stages (4–5 cell embryos, n = 24; 6–7 cell embryos, n = 7; 8 cell embryos, n = 9).
*Figure 3 continued on next page*

*Figure 3 continued*

(C) Maximal projections of four time-frames during live imaging of embryos from the CBF1-VENUS reporter line. Time since the onset of time lapse is indicated. (D) 3D reconstruction of the time-lapse imaging of a representative embryo. A selected cell and its progeny are highlighted in orange. Blue blastomeres indicate inner position and gray blastomeres indicate outer position. (E) Mean intensity levels of VENUS in all the families of the live imaged embryos (n = 7) according to the position of a cell and their progeny in the first and the last time frame. (F) Initial intensity levels of VENUS in all the families of the live imaged embryos according to the position of a cell and their progeny in the first and the last time frame. For (E) and (F), n = 13 families for IN-ICM, n = 11 families for IN–TE + ICM, n = 55 families for OUT TE, n = 16 families for OUT–TE + ICM. ***p<0.001, *p<0.05 by ANOVA with Bonferroni post-test.

DOI: https://doi.org/10.7554/eLife.42930.011

The following figure supplements are available for figure 3:

**Figure supplement 1.** CBF1-VENUS activity in embryos before compaction.
DOI: https://doi.org/10.7554/eLife.42930.012

**Figure supplement 2.** CBF1-VENUS dynamics in morula to blastocyst transition.
DOI: https://doi.org/10.7554/eLife.42930.013

their descendant cells. We next classified families according to the position of the cells in the first and final time points. This allowed us to divide the cells in four groups: 'IN-ICM' (cells that began in an inner position and their descendants remained in an inner position), 'IN-TE +ICM' (cells that began in an inner position and at least one of their descendants ended up in an inner position but other/s in an outer position), 'OUT-TE' (cells that began in an outer position and their descendants remained in an outer position), and 'OUT-TE +ICM' (cells that began in an outer position and at least one of their descendants ended up in an outer position but other/s ended up in an inner position).

Confirming previous findings (*Anani et al., 2014*; *McDole and Zheng, 2012*; *Posfai et al., 2017*; *Toyooka et al., 2016*; *Watanabe et al., 2014*), although most of the cells of the blastocyst retain the position of their predecessor cell in the compacted morula, a small percentage change their location (*Figure 3—figure supplement 2B*). We next measured intensity levels of the reporter in all cells within families, and determined if it correlated with their position during the time lapse. Notch activity levels were variable among families and embryos, but we detected higher and increasing levels in OUT-TE families while IN-ICM families generally showed lower and decreasing levels (*Figure 3—figure supplement 2C–D*). The intensity levels in families that contributed both inner and outer cells did not follow a clear pattern (*Figure 3—figure supplement 2E*). When we analysed the mean intensity for each group, we saw that VENUS levels were significantly lower in the families that were always inside as compared to the families that were always outside (*Figure 3E*). Interestingly, this difference was already manifest when we measured the initial intensity in the first time point (*Figure 3F*). In the families whose cells end up in both inner and outer position, VENUS levels were intermediate (*Figure 3E,F*).

Therefore, the analysis of the CBF1-VENUS line showed that the reporter is active before the first lineage decision is taken, and that differences in the levels of pathway activation in inner or outer cells of the compacted morula correlate with the final position of their descendants in the blastocyst.

## Different Notch levels influences cell position in the morula and the blastocyst

We have previously shown that increasing the activity of the Notch signalling pathway leads to a preferential allocation of cells to the outer trophectoderm of the blastocyst (*Rayon et al., 2014*). However, we had not tested the onset of

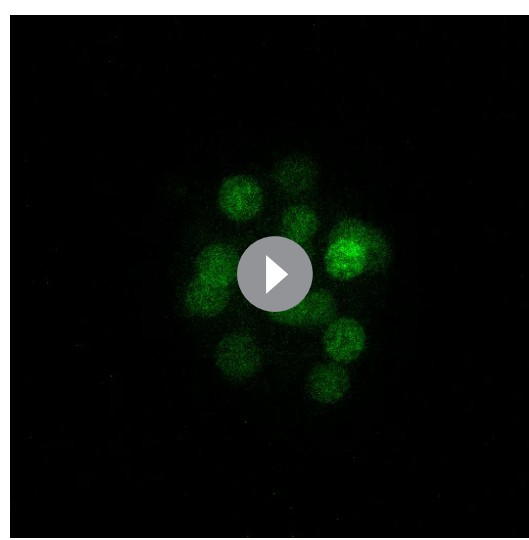

**Video 1.** Time lapse imaging of a mouse embryo from the CBF1-VENUS line during morula to blastocyst transition.

DOI: https://doi.org/10.7554/eLife.42930.014

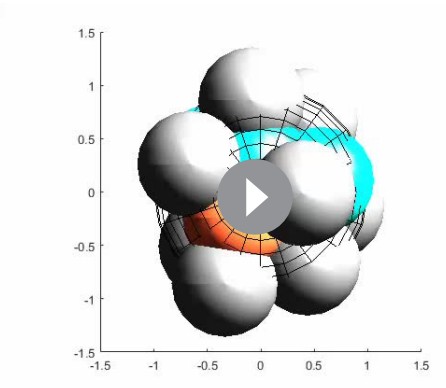

**Video 2.** 3D reconstruction of a mouse embryo from the CBF1-VENUS line after live imaging during morula to blastocyst transition.
DOI: https://doi.org/10.7554/eLife.42930.015

this effect and whether blocking Notch would have an effect in early embryos. To address these questions, we used a genetic mosaic line (iChr-Notch-Mosaic) that allowed us to generate cells with different Notch activity levels within the same embryo (*Pontes-Quero et al., 2017*). The construct consists of three different cassettes preceded by a specific LoxP site. The first cassette is a H2B-CHERRY fluorescent protein and generates wildtype cells. The second cassette contains a dominant-negative version of *Mastermind-like 1* (DN-MAML1), a transcriptional co-activator of the Notch pathway, linked to a H2B-eGFP by a cleavable 2A peptide, whose expression leads to the loss of function (LOF) of the pathway, while the third is a gain of function (GOF) cassette through the expression of a constitutively active NICD linked to an HA-H2B-Cerulean (*Figure 4A*). The specific LoxP sites are mutually exclusive, so in any unique cell there will be only one possible outcome as the result of Cre-mediated recombination. We used a Polr2a$^{CreERT2}$ driver which is ubiquitously expressed and inducible by tamoxifen (*Guerra et al., 2003*). We induced recombination by adding 4OH-Tx (4-hydroxy-tamoxifen) from the 2- to the 4-cell stage, aiming to achieve a situation where cells expressing each cassette derive from a single recombined blastomere, and we evaluated recombination in the late morula (<32 cells) or in the blastocyst (*Figure 4A*). We performed immunofluorescent assays with three antibodies to distinguish the three cassettes. The wildtype cassette was detected by an anti-RFP antibody, the LOF by an anti-GFP antibody, and the GOF an anti-HA antibody. However, GOF cells were triple positive because of cross-reactivity between antibodies and the HA-H2B-Cerulean protein (*Figure 4A,B*). To validate this strategy, we induced recombination in a iChr-Notch-Mosaic ES cell line, and sorted cells according to their fluorescence. Testing the antibodies described above in these populations confirmed that they correctly identified cells derived from each different recombination event (*Figure 4—figure supplement 1A*). Confirmation that we were modifying Notch signalling as predicted with the LOF and GOF cassettes came from analysing nuclear CDX2 levels, as a readout of Notch activity. Notch LOF blastomeres showed lower CDX2 levels than wildtype and GOF, and these higher levels than wildtypes (*Figure 4—figure supplement 1B*).

We selected embryos in which at least two recombination events leading to LOF and GOF cells had occurred, and analysed the percentage of unrecombined cells (36%), and cells expressing the control, LOF or GOF cassette. Although the probabilities of recombination are higher when the LoxP sites are closer to one another (the control recombination event in this case), we found that most of the recombined cells were Notch GOF (36% of total cells) while only a small proportion (11%) were Notch LOF, (*Figure 4C*; *Figure 4—figure supplement 2A,B*). If we used a control line (iChr-Control-Mosaic; *Pontes-Quero et al., 2017*) carrying the same construct and reporters but not the LOF or GOF cassettes (*Figure 4—figure supplement 3A*), we observed a similar proportion of unrecombined cells (34%) but in this case the most probable event (red cells) was the most abundant (34%), as expected (*Figure 4—figure supplement 3B*). These results suggest that Notch activity could differentially affect cell proliferation or cell loss in the embryo.

Next, we determined the proportion of cells from each population that were in an inner or outer position. Approximately 60% of unrecombined or wild type (red) cells, both being controls, were located at outer positions in both morula and blastocyst stage. However, Notch-LOF cells (green) were enriched at inner positons of the morula or more clearly in the inner cell mas of the blastocyst, while Notch-GOF cells (blue) tended to occupy outer positions (*Figure 4D,E*). These differences were not observed when we analysed embryos from the iChr-Control-Mosaic line (*Figure 4—figure supplement 3C*). These experiments show how manipulating levels of Notch pathway activity as early as the 4-cell stage instructs cells to adopt an inner or outer position at later stages.

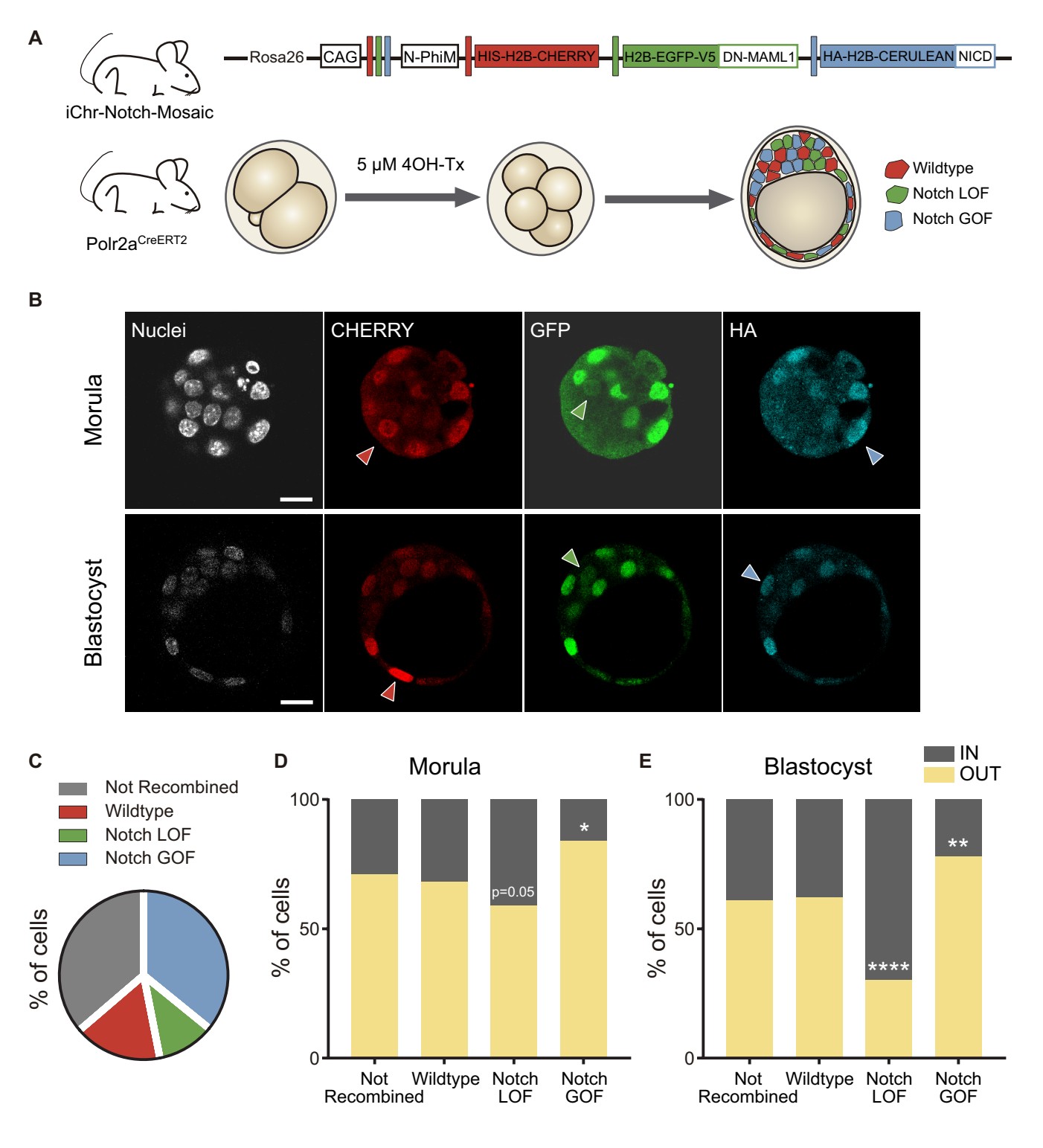

**Figure 4.** Differences in Notch activity drive cell fate in the preimplantation embryo. (**A**) Schematic diagram of the experimental strategy, where iChr-Notch-Mosaic mice were crossed with Polr2a$^{CreERT2}$ driver. Embryos were collected and treated with 4OH-Tamoxifen from 2- to 4-cell stage to induce recombination. At morula and blastocyst stage, embryos were fixed and immunostained. (**B**) Optical section of confocal images after immunostaining for RFP, GFP and HA. Arrowheads indicate examples of cells recombined for the wildtype cassette (red), the Notch loss of function cassette (LOF, green) or the Notch gain of function cassette (GOF, blue). Nuclei were stained with DAPI. Scale bars, 20 μm. (**C**) Percentage of not recombined cells or recombined for each cassette in morulae (n = 10) and blastocysts (n = 11). (**D–E**) Percentage of not recombined cells or recombined for each cassette

*Figure 4 continued on next page*

*Figure 4 continued*

that are in an inner or outer position at the morula (D; not recombined, n = 69; wildtype, n = 33; LOF, n = 29; GOF, n = 83) or blastocyst stage (E; not recombined, n = 190; wildtype, n = 87; LOF, n = 50; GOF, n = 173). *p<0.05, **p<0.01, ****p<0.0001 in relation to not recombined cells by Chi-square test.

DOI: https://doi.org/10.7554/eLife.42930.016

The following figure supplements are available for figure 4:

**Figure supplement 1.** Differences in Notch activity correlate with CDX2 levels.

DOI: https://doi.org/10.7554/eLife.42930.017

**Figure supplement 2.** Confronting Notch activity levels in the preimplantation embryo.

DOI: https://doi.org/10.7554/eLife.42930.018

**Figure supplement 3.** Distribution of recombination events in a control mosaic mouse line.

DOI: https://doi.org/10.7554/eLife.42930.019

## Lack of *Rbpj* represses *Tle4* and *Tbx3*, and disrupts the triggering of differentiation programs in the early embryo

Results described above show that the Notch pathway plays an early role in mouse development, non-redundant with that of the Hippo pathway, in regulating *Cdx2* gene expression and in determining the position of cells to inner or outer locations. To gain further insight into how Notch is acting during preimplantation development, we carried out RNA-sequencing (RNA-seq) in control and *Rbpj*^-/- single morulae. 2028 genes were differentially expressed (*Figure 5—source data 1*), close to 70% of which were downregulated suggesting that *Rbpj* is mainly activating gene expression at the morula stage.

Among the downregulated genes we found *Cdx2* and other TE associated genes such as *Gata3* or *Fgfr2* (*Haffner-Krausz et al., 1999*; *Home et al., 2017*; *Home et al., 2009*; *Ralston et al., 2010*); genes related with the Hippo pathway (*Nf2*, *Amotl2*, *Lats2*) and, interestingly, also genes related with the embryonic pluripotency network including *Sall1*, *Sall4*, *Tbx3* or *Sox21* (*Goolam et al., 2016*; *Han et al., 2010*; *Karantzali et al., 2011*; *Niwa et al., 2009*; *Yang et al., 2010*) (*Figure 5A*, *Figure 5—figure supplement 1A*). Among the upregulated genes, we found *Dppa3* (*Stella*) and *Prdm14*, which have been characterised as naïve pluripotency markers(*Hayashi et al., 2008*; *Yamaji et al., 2013*). In addition, a large set of chromatin modifiers were differentially expressed (*Figure 5—figure supplement 1B*). Important chromatin dynamics have been reported during preimplantation development (*Burton and Torres-Padilla, 2014*), which could fit in with the broad misregulation of transcription in the mutant embryos. Remarkably, some of the downregulated modifiers like *Dnmt3b* or *Kdm6a* have been shown to be enriched in TE conversely to *Prdm14* (*Burton et al., 2013*). Overall, the transcriptome profiling suggests that embryos lacking *Rbpj* do not properly trigger trophectoderm differentiation programs, and that they also affect pluripotency related genes.

To identify direct targets of Notch signalling at this stage, we searched for putative RBPJ binding sites in the vicinity of differentially expressed genes. We established an arbitrary window of 10 Kb surrounding each gene to perform the analysis and found RBPJ binding motifs in 1487 genes. We then examined how many of these putative binding sites were located in regions of open chromatin, a hallmark for active regulatory elements. For this, we took advantage of ATAC-seq profiles from published datasets of 8 cell mouse embryos (*Wu et al., 2016*), and reduced our list to 282 genes (*Figure 5B*; *Figure 5—source data 2*). Among these was *Cdx2*, where the predicted RBPJ sites and ATAC-seq open chromatin signature mapped to the TEE enhancer we had previously characterised (*Rayon et al., 2014*), thus validating this approach.

We selected two genes as putative Notch targets, that were downregulated in *Rbpj*^-/- morulae and had been previously associated with exit from pluripotency in mouse ES cells: those coding for the Groucho-family transcriptional repressor TLE4 (*Laing et al., 2015*), and the T-box family transcription factor TBX3 (*Russell et al., 2015*; *Waghray et al., 2015*). Both genes are heterogeneously expressed in ES cells and repress naïve pluripotency genes. We hypothesized that *Tle4* and *Tbx3* could be direct targets of Notch, and that their downregulation could in part explain the blockade in differentiation that we observe in the RNA-seq. We independently confirmed downregulation of

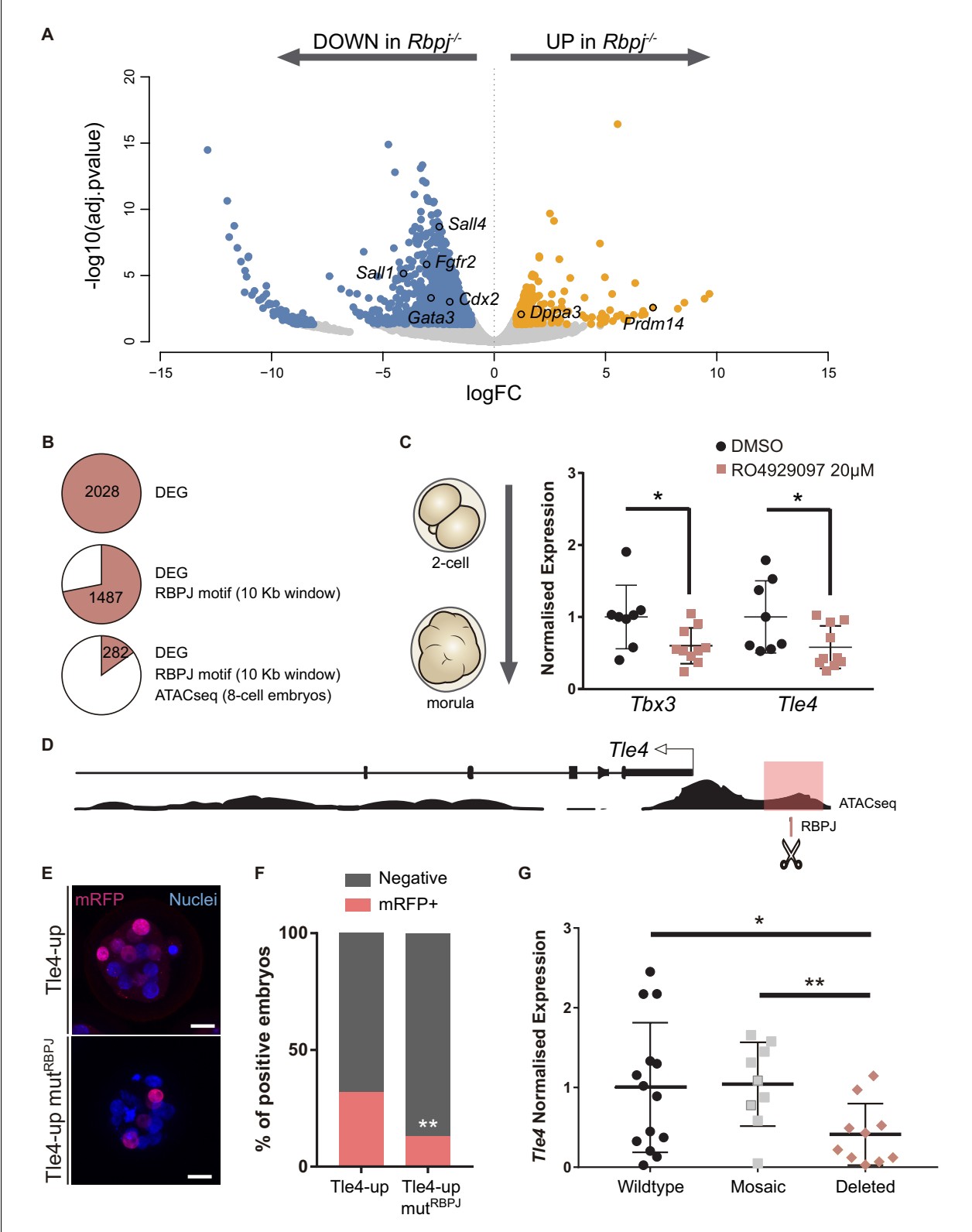

**Figure 5.** *Tle4* is a direct transcriptional target of Notch. (**A**) Volcano plot of differentially expressed genes between wildtype and *Rbpj*[-/-] single morulae. In blue, genes downregulated in *Rbpj*[-/-] (adj.pvalue <0.05 and logFC < −1); in orange, genes upregulated in *Rbpj*[-/-] (adj.pvalue <0.05 and logFC >1). Representative genes are indicated. (**B**) Number of differentially expressed genes (DEG) between wildtype and *Rbpj*[-/-] morulae (top), those that have a RBPJ motif in a 10 Kb window surrounding the gene (middle), and those in which this site is included in an open chromatin ATAC-seq peak

*Figure 5 continued on next page*

*Figure 5 continued*

in 8 cell embryos (*Wu et al., 2016*) (bottom). (C) *Tbx3* and *Tle4* normalised expression in pools of 25 embryos after treatment with RO4929097 to block Notch from 2 cell to morula stage. Pools of embryos treated with DMSO were used as controls (n = 8–10). n represents number of unique pools of 25 embryos. (D) Genomic landscape of the region upstream *Tle4* indicating the location of the RBPJ motif and the ATAC-seq track. (E) Maximal projection confocal images after RFP immunostaining of representative transgenic embryos for the region highlighted in pink in (D) (top) or the mutated version for the RBPJ site (bottom). Nuclei were stained with DAPI. Scale bar, 20 μm. (F) Percentage of positive embryos in the transient transgenic assay of Tle4-up region (n = 137) or the mutated version (n = 169). (G) *Tle4* normalised expression in single embryos (wildtype, n = 14; mosaic, n = 9; deleted, n = 10) after CRISPR/Cas9 deletion of the region containing the RBPJ motif. Data are means ± s.d. *p<0.05 by Student's t test in (C) and (G). **p<0.01 by Student's t test in (G) or by Fisher's exact test in (F).
DOI: https://doi.org/10.7554/eLife.42930.020

The following source data and figure supplements are available for figure 5:

**Source data 1.** Table with differentially expressed genes (DEG) obtained after the RNA-seq analysis in control and *Rbpj* mutant morulae.
DOI: https://doi.org/10.7554/eLife.42930.023

**Source data 2.** Table with differentially expressed genes (DEG) that have an RBPJ consensus binding site included in an open chromatin ATAC-seq peak in a 10 Kb window surrounding the gene.
DOI: https://doi.org/10.7554/eLife.42930.024

**Figure supplement 1.** Transcriptome analysis of *Rbpj*⁻ᐟ⁻ single morulae.
DOI: https://doi.org/10.7554/eLife.42930.021

**Figure supplement 2.** Regulation of *Tbx3* in the morula and genome editing.
DOI: https://doi.org/10.7554/eLife.42930.022

their expression after blocking the Notch pathway by treating wildtype embryos with the RO4929097 inhibitor from 2 cell to morula stage (*Figure 5C*).

An RBPJ motif search within ATAC-seq peaks in the vicinity of the genes identified two potential candidate regions located 1.3 Kb upstream of *Tle4* (Tle4-up; *Figure 5D*) and in the seventh intron of *Tbx3* (Tbx3-i7; *Figure 5—figure supplement 2A*), respectively. By means of transient transgenic assays (*Rayon et al., 2014*), we proved that these regions could act as transcriptional enhancers driving H2B-mRFP reporter expression in the morula (32% positive embryos for the 700 bp Tle4-up element, *Figure 5E,F*; and 56% for the 600 bp Tbx3-i7 element; *Figure 5—figure supplement 2B,C*). To test if Notch was directly involved, we mutated the RBPJ motif inside these regions and found that the activity of the Tle4-up mut^RBPJ fragment was significantly diminished (from 32% to 13% positive embryos, *Figure 5E,F*) while the Tbx3-i7 mut^RBPJ fragment was not affected (60% positive embryos, *Figure 5—figure supplement 2B–C*). Finally, to examine whether these enhancers were necessary for the expression of their putative target genes, we deleted the regions within the enhancers that contained the RBPJ motif by CRISPR/Cas9 mediated genome editing (*Ran et al., 2013*), and analysed gene expression by qPCR on individually edited morulae. For this we used two guide-RNAs flanking the regions that contained the putative RBPJ binding site, what generated deletions of approximately 150 bp (*Figure 5—figure supplement 2E,F*). We observed a significant decrease in *Tle4* expression in edited embryos (deleted, n = 10) as compared to injected embryos that had been partially (mosaic, n = 9) or not (wildtype, n = 14) edited (*Figure 5G*). However, *Tbx3* expression did not change when the RBPJ motif from the seventh intron was deleted (*Figure 5—figure supplement 2D*). These assays provide evidence that these genomic regions act as *cis*-regulatory elements and, in the case of *Tle4*, are directly regulated by RBPJ and necessary for correct expression.

## Notch levels coordinate the balance between naïve pluripotency and triggering of differentiation in ES cells

The transcriptomic profiling carried out in *Rbpj*⁻ᐟ⁻ embryos identified genes related with naïve pluripotency among the upregulated genes. Naïve pluripotency corresponds to a state in which cells are not prone to differentiate, in contrast to primed pluripotency (*Kalkan and Smith, 2014*). These pluripotent states as well as the transition between them have been extensively studied in ES cells and EpiLCs, in vitro counterparts of the epiblast of the blastocyst stage preimplantation embryo and the postimplantation pre-gastrulating epiblast respectively (*Hackett and Surani, 2014*). Interestingly, some of these naïve markers such as *Prdm14* are initially expressed at the 2- and 4-cell stage, switched off in the morula and re-expressed in the ICM of the blastocyst (*Burton et al., 2013*).

Analysis of published single-cell RNA-seq data (*Goolam et al., 2016*) confirmed that *Prdm14* decreased dramatically from the 4- to 8-cell stage, and expression of *Dppa3* also decreases from the 2- to 4-cell stage (*Figure 6—figure supplement 1*). In contrast, *Tle4* and *Tbx3* levels increased from the 4- to 8-cell stage (*Figure 6—figure supplement 1*). Our data from *Rbpj*⁻/⁻ morulae suggests that embryos do not switch off *Prdm14* and *Dppa3*, and inhibiting Notch with RO4929097 from the 2- to 4-cell stage confirmed the effect on *Prdm14*, whose levels were significantly increased after the treatment (*Figure 6—figure supplement 2*).

We wondered if the effect of Notch guiding differentiation programs that we had seen in the embryo was also occurring in ES cells. We used iChr-Notch-Mosaic ES cells (*Pontes-Quero et al., 2017*) to confront populations with different Notch levels using the same strategy than we had previously used in the embryo (*Figure 6A*). After recombination by transfection with *Cre*, ES cells were sorted according to the fluorescent reporter cassette they expressed (*Figure 6B*). We measured expression levels of naïve pluripotency markers by qPCR, and found that levels of *Prdm14* and *Dppa3* correlated negatively with Notch activity but other markers such as *Nanog* or *Esrrb* were not altered (*Figure 6C*). We next asked how Notch would affect the differentiation potential of pluripotent cells using this system. For that, we allowed sorted iChr-Notch-Mosaic ES cells grown in serum +LIF to differentiate for 48 hr after LIF removal and analysed the expression of genes related to early differentiation at different time points (*Figure 6D*). On the one hand, we observed that the peak of expression of *Tle4*, and the early epiblast markers *Fgf5* and *Pou3f1* occurred earlier and remained at higher levels in Notch GOF than in wildtype ES cells. On the other hand, Notch LOF cells never reached normal levels of *Tbx3* or *Fgf5* during the differentiation process (*Figure 6E*). These results suggest that Notch is not only sufficient to drive expression of some differentiation markers such as *Tle4*, but also necessary to achieve proper levels of others such as *Tbx3*. However, modulation of Notch levels is not enough to change expression of pluripotency markers once ES cells have started the differentiation process (*Figure 6—figure supplement 3*). If we carried out the experiment but using ES cells maintained under naïve conditions (2i + LIF), we observed similar dynamics for *Tle4* and *Tbx3* and other early epiblast markers, but no changes in later differentiating genes (*Figure 6—figure supplement 4*). Overall, our results suggest that Notch is involved in coordinating exit from pluripotency and promoting cell differentiation in ES cells, mirroring its role in the early embryo.

## Discussion

During the first three days of mouse embryonic development, cells lose their totipotent capacity as they form the first differentiated population, the trophectoderm (TE). In this study, we show that Notch signalling regulates the early expression of *Cdx2*, a key element in TE specification, and that this is later reinforced by the input of Hippo signalling through YAP and TEAD4. Hippo has been shown to act as a readout of cell polarity (*Anani et al., 2014*; *Hirate et al., 2015*) and it activates *Cdx2* in cells that have established an apical domain. However, the initial triggering of *Cdx2* both in inner and outer cells (*Dietrich and Hiiragi, 2007*; *Posfai et al., 2017*) suggested that inputs other than Hippo would initially be acting because its expression could not be explained only by YAP/TEAD4 activity. In fact, previous reports have described that although in most *Tead4*⁻/⁻ blastocysts CDX2 is not detected, earlier *Tead4*⁻/⁻ morulae retain CDX2 expression (*Nishioka et al., 2008*). Similarly, double *Wwtr1;Yap1* mutant embryos show some residual CDX2 expression (*Frum et al., 2018*). In agreement with these observations, we found blastomeres in the morula that express CDX2 but do not have nuclear YAP. In this situation, expression of CDX2 is likely due to Notch activity as the CBF1-VENUS reporter, used as a proxy for activity of the pathway (*Nowotschin et al., 2013*), is present in those cells. The analysis of *Rbpj* and *Notch1* mutants in early and late morulae, as well as pharmacological treatments of preimplantation embryos, further support the notion that the input provided by Notch is necessary for the early phases of *Cdx2* expression. These results show that Notch and Hippo have non-redundant but partially overlapping roles in early and late phases of *Cdx2* expression, respectively. Furthermore, only double knockout morulae for *Rbpj* and *Tead4* completely lack CDX2, and all CDX2 positive cells have at least one of the two pathways active. These findings support a model whereby overlapping or complementary inputs from different signalling pathways may provide robustness in the system, buffering any disturbances and ensuring

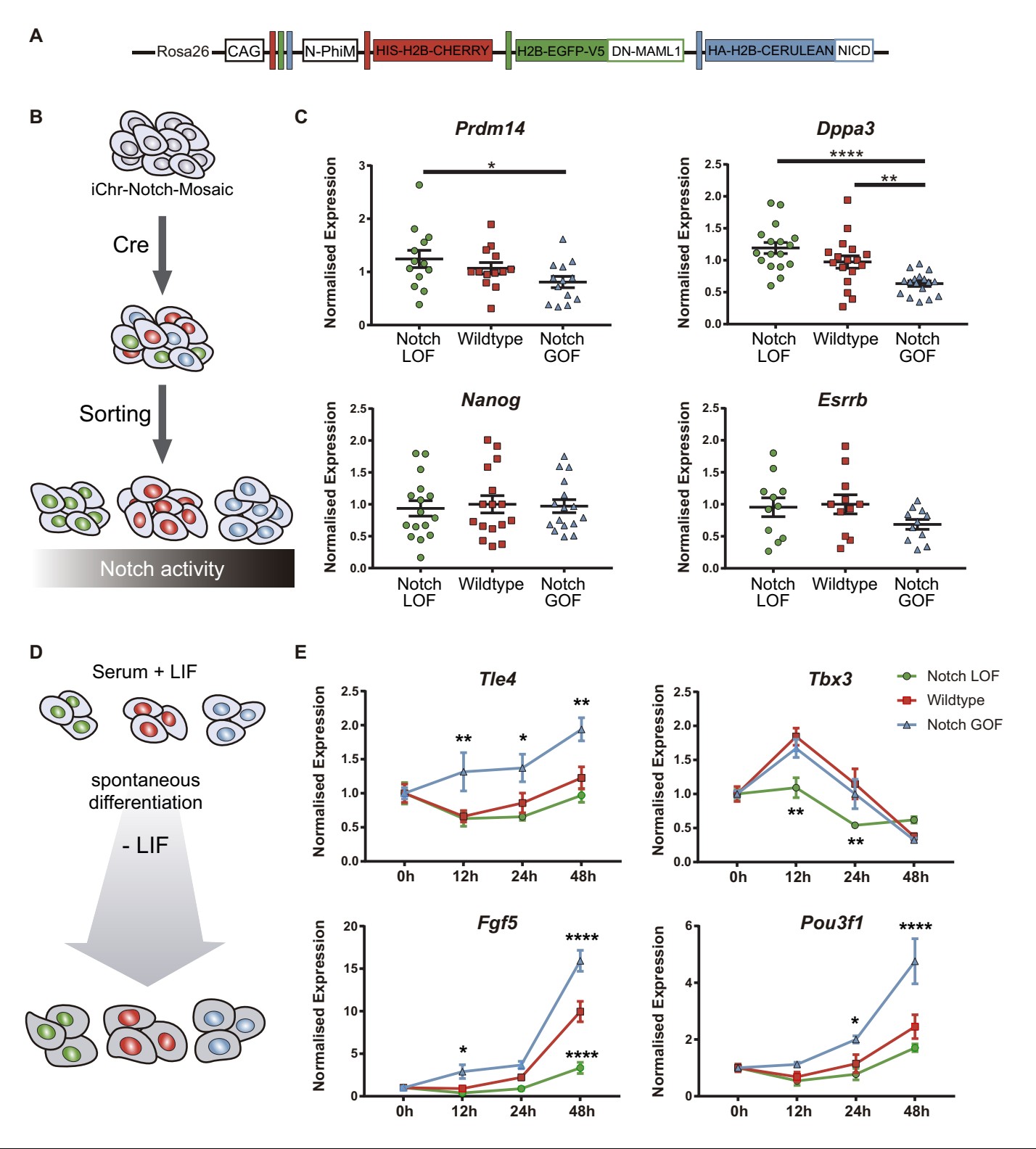

**Figure 6.** Notch promotes exit from naïve pluripotency and cell differentiation of ES cells. (**A**) Construct showing the wildtype (red), Notch loss of function (green) and Notch gain of function (blue) cassettes from iChr-Notch-Mosaic ES cells. (**B**) Schematic diagram of experimental design, where iChr-Notch-Mosaic ES cells were recombined with Cre and sorted according to Notch activity. (**C**) *Prdm14*, *Dppa3*, *Nanog* and *Esrrb* normalised expression in iChr-Notch-Mosaic ESCs after sorting of Notch LOF, wildtype and Notch GOF populations (n = 13 for *Prdm14*, n = 17 for *Dppa3*, n = 16

*Figure 6 continued on next page*

*Figure 6 continued*

for *Nanog*, n = 11 for *Esrrb*). (**D**) Schematic diagram of experimental design, where sorted recombined iChr-Notch-Mosaic ES cells were differentiated after LIF removal. (**E**) *Tle4*, *Tbx3*, *Fgf5* and *Pou3f1* normalised expression in Notch LOF, Wildtype and Notch GOF cells at 0 hr, 12 hr, 24 hr and 48 hr after LIF withdrawal (n = 6). Data are means ± s.e.m. *p<0.05, **p<0.01, ***p<0.001, ****p<0.0001 in relation to wildtype cells by ANOVA with Fisher post-test.

DOI: https://doi.org/10.7554/eLife.42930.025

The following figure supplements are available for figure 6:

**Figure supplement 1.** Expression of naïve pluripotency markers in the preimplantation embryo.

DOI: https://doi.org/10.7554/eLife.42930.026

**Figure supplement 2.** Notch inhibition increases *Prdm14* in the 4 cell embryo.

DOI: https://doi.org/10.7554/eLife.42930.027

**Figure supplement 3.** Expression of naïve pluripotency markers in differentiating ES cells.

DOI: https://doi.org/10.7554/eLife.42930.028

**Figure supplement 4.** Notch promotes differentiation of naïve ES cells.

DOI: https://doi.org/10.7554/eLife.42930.029

proper development (*Menchero et al., 2017*). In such a model, Notch and Hippo would ensure the correct specification and maintenance of the TE respectively (*Rayon et al., 2014*).

The crosstalk between YAP and Notch has been studied in different cellular contexts (*Totaro et al., 2018*). YAP acts upstream of Notch in controlling epidermal stem cell fate or liver cell fate (*Totaro et al., 2017*; *Yimlamai et al., 2014*) while Notch is upstream of YAP in the corneal epithelium during chronic inflammation (*Nowell et al., 2016*). Also, YAP and Notch can cooperate to control the onset of oscillations in the segmentation clock (*Hubaud et al., 2017*) and they interact to promote the expression of *Jag1* in smooth muscle cells (*Manderfield et al., 2015*). During TE establishment, YAP and Notch have also been shown to interact through SBNO1, and act synergistically to regulate *Cdx2* (*Watanabe et al., 2017*). In this context, our results show that both pathways are acting in parallel since there is no correlation among YAP and CBF1-VENUS expression levels in single blastomeres in morula stage embryos. In addition, loss of the NOTCH1 receptor or RBPJ does not affect YAP/TEAD4 localisation and vice versa, *Tead4* knockout does not alter CBF-VENUS expression in the blastocyst. Nevertheless, several components of the Hippo pathway are downregulated in *Rbpj*[-/-] morulae, so we cannot rule out the possibility of cross-transcriptional regulation between the pathways.

The role of Notch signalling in the specification of cell fates during development has been widely studied (*Koch et al., 2013*). Notch promotes heterogeneities and reinforces differences between neighbouring cells, explaining the segregation of cell fates in multiple processes and in different species (*Artavanis-Tsakonas et al., 1999*). The heterogeneous activity of CBF-VENUS in the 4-cell stage coincides with the loss of cell equivalence and emergence of differences among blastomeres. Other factors have been shown to be differentially expressed among blastomeres of the 4 cell mouse embryo (*Burton et al., 2013*; *Goolam et al., 2016*), suggesting that this is the moment when cells lose their homogeneous state to start desynchronizing and differentiating. Interestingly, *Prdm14*, one of these factors, and Notch show divergent patterns of expression during development. *Prdm14* is first expressed at the 2- and 4-cell stage, to be turned off and then re-expressed in the ICM of the blastocyst and later in the primordial germ cells (*Burton et al., 2013*; *Yamaji et al., 2008*). In contrast, the Notch pathway, as revealed by the CBF-Venus reporter, begins to be active at the 4-cell stage, it is active in most of the cells of the morula, and is later restricted to the TE of the blastocyst. After implantation, Notch activity is detected throughout the epiblast (*Nowotschin et al., 2013*). It has been suggested that *Prdm14* expression coincides with conditions where groups of cells show an undetermined state, while Notch is activated when cells transition towards their next developmental phase. Our results suggest that Notch would be regulating these transitions by downregulating *Prdm14* expression. In line with the upregulation of *Prdm14* in embryos that lack Notch activity, we observed in the RNA-seq data from *Rbpj*[-/-] morulae a downregulation of Fgf receptors (*Fgfr1*, *2* and *3*) and DNA methyltransferases (*Dnmt3b*, *Dnmt1*), which are known to be repressed by PRDM14 (*Grabole et al., 2013*; *Yamaji et al., 2013*). It is also interesting to note that in our mosaic ES cell experiments, Notch levels correlate with those of *Prdm14* and *Dppa3*, but not with other pluripotency markers such as *Nanog* or *Esrrb*. Therefore, Notch is not

simply turning off the general pluripotency network to promote differentiation, but acting on a subset of early naïve pluripotency markers.

Interplay between Notch and chromatin remodellers has been reported in several situations (*Schwanbeck, 2015*). Expression changes in chromatin modifiers precede the action of transcription factors that consolidate lineage choices during preimplantation development (*Burton et al., 2013*). Therefore, these alterations suggest that $Rbpj^{-/-}$ embryos do not established correct epigenetic landscapes, do not switch off early markers such as *Prdm14* or *Dppa3* and are not able to properly trigger differentiation programs leading to a delay in the expression of lineage specifiers such as *Cdx2*. In this regard, it is interesting to note that *Rbpj* mutant morulae downregulate *Chaf1a*, which encodes the large subunit of the histone-chaperone CAF-1. Loss of CAF-1 promotes ES cells to transit to an earlier, totipotent 2-cell-like state (*Ishiuchi et al., 2015*), and acts as a barrier for reprogramming (*Cheloufi et al., 2015*). Furthermore, knockout of *Chaf1a* leads to developmental arrest at the 16-cell stage and a loss of heterochromatin (*Houlard et al., 2006*). Thus, CAF-1 acts as a driver of differentiation in pluripotent cells. Interestingly, studies in *Drosophila* have shown that CAF-1 mediates downstream effect of the Notch pathway (*Yu et al., 2013*). On the other hand, *Asf1a*, which encodes another histone chaperone, is among the few genes observed to be upregulated in $Rbpj^{-/-}$ embryos. Forced expression of *Asf1a* promotes reprogramming of human ES cells (*Gonzalez-Muñoz et al., 2014*), revealing a critical role in maintaining pluripotency. Furthermore, *Suv39h1*, a regulator of H3K9me3-heterochromatin that restrict cell plasticity and stemness (*Yadav et al., 2018*), is also downregulated in Notch loss-of-function morulae. In conclusion, we observed that during preimplantation development, Notch regulates critical epigenetic components that mediate transitions along the progressive restriction of potency that occurs in the early embryo.

In this study, we have also identified novel putative targets positively regulated by the Notch pathway, such as *Tle4* and *Tbx3* whose role in the exit from pluripotency has been described in ES cells (*Laing et al., 2015*; *Russell et al., 2015*; *Waghray et al., 2015*). Their increase in expression from 2 cell to morula supports their possible role in promoting early differentiation in vivo as well. TLE4 does not bind directly to DNA, but associates with other proteins to act as a transcriptional corepressor (*Kaul et al., 2015*). It will be of great interest to identify its transcriptional partners during preimplantation development and elucidate the mechanism by which it allows cell differentiation in this context. The role of TBX3 is more complex since, in addition to promoting differentiation, it has also been associated with pluripotency maintenance (*Han et al., 2010*; *Niwa et al., 2009*). Furthermore, in vivo TBX3 is detected in most of the cells of the morula but it is later restricted to the ICM (*Russell et al., 2015*), following a complementary pattern to Notch. Thus, *Tbx3* regulation must involve Notch-dependant and Notch-independent inputs, what could explain why the mutation or deletion of the RBPJ motif present in the intronic *Tbx3* regulatory element did not disrupt enhancer activity or endogenous expression.

The role of Notch in ES cells had already been explored in the context of neural differentiation (*Lowell et al., 2006*). Blocking Notch signalling prevents ES cells from adopting a neural fate while its overexpression increases the frequency of neural specification. Our results suggest that Notch might have a more general role in promoting early differentiation, with a more specific function in neural specification at later stages (*Lowell et al., 2006*). In summary, our findings suggest that Notch acts by promoting the gradual loss of potency in the early embryo, which is subsequently reinforced by additional mechanisms, such as heterochromatin formation before the morula stage, or differential activation of the Hippo pathway at the morula-to-blastocyst transition. Therefore, in order to correctly specify a given lineage, such as the trophectoderm, Notch is simultaneously activating fate choice markers such as *Cdx2* and inducing a differentiation-prone state by lowering levels of naïve markers.

# Materials and methods

**Key resources table**

| Reagent type (species) or resource | Designation | Source or reference | Identifiers | Additional information |
|---|---|---|---|---|

*Continued on next page*

*Continued*

| Reagent type (species) or resource | Designation | Source or reference | Identifiers | Additional information |
|---|---|---|---|---|
| Strain (*Mus musculus*) | CD1 | Charles Rivers | | |
| Strain (*Mus musculus*) | C57Bl/6 | Charles Rivers | | |
| Strain (*Mus musculus*) | CBA | Charles Rivers | | |
| Genetic reagent (*M. musculus*) | CBF1-VENUS | *Nowotschin et al., 2013* | MGI:5487911 | Dr. Anna-Katerina Hadjantonakis |
| Genetic reagent (*M. musculus*) | Rbpj null | *Oka et al., 1995* | MGI:1857411 | Dr. Jose Luis de la Pompa |
| Genetic reagent (*M. musculus*) | Notch1 null | *Conlon et al., 1995* | MGI:1857230 | Dr. Jose Luis de la Pompa |
| Genetic reagent (*M. musculus*) | Tead4 null | *Nishioka et al., 2008* | MGI:3770620 | Dr. Hiroshi Sasaki |
| Genetic reagent (*M. musculus*) | iChr-Control-Mosaic | *Pontes-Quero et al., 2017* | MGI:6108166 | Dr. Rui Benedito |
| Genetic reagent (*M. musculus*) | iChr-Notch-Mosaic | *Pontes-Quero et al., 2017* | | Dr. Rui Benedito |
| Genetic reagent (*M. musculus*) | Polr2a-CreERT2 | *Guerra et al., 2003* | MGI:3772332 | Dr. Miguel Torres |
| Cell line (*M. musculus*) | iChr-Notch-Mosaic ESC | *Pontes-Quero et al., 2017* | | Dr. Rui Benedito |
| Antibody | anti-CDX2 (mouse monoclonal) | BioGenex | MU392-UC | (1:200) |
| Antibody | anti-CDX2 (rabbit monoclonal) | Abcam | ab76541 | (1:200) |
| Antibody | anti-YAP (mouse monoclonal) | Santa Cruz Biotechnology | sc-101199 | (1:200) |
| Antibody | anti-pERM (rabbit polyclonal) | Cell Signalling | 3141 | (1:250) |
| Antibody | anti-E-Cadherin (rat monoclonal) | Sigma | U3254 | (1:250) |
| Antibody | anti-TEAD4 (mouse monoclonal) | Abcam | ab58310 | (1:100) |
| Antibody | anti-DsRed (rabbit polyclonal) | Living Colors, Clontech | 632496 | (1:400) |
| Antibody | anti-GFP (goat polyclonal) | Acris, Origene | R1091P | (1:200) |
| Antibody | anti-HA (rat monoclonal) | Sigma | 11867423001 | (1:200) |
| Recombinant DNA reagent | Tle4-up_H2BmRFP | This paper | | modified pBluescript vector |
| Recombinant DNA reagent | Tbx3-i7_H2BmRFP | This paper | | modified pBluescript vector |

*Continued on next page*

*Continued*

| Reagent type (species) or resource | Designation | Source or reference | Identifiers | Additional information |
|---|---|---|---|---|
| Sequence-based reagent | qPCR primers | This paper | | See *Supplementary file 1* |
| Commercial assay or kit | PicoPure RNA Isolation Kit | ThermoFisher | KIT0204 | |
| Chemical compound, drug | RO4929097 | Selleckchem | S1575 | |
| Chemical compound, drug | Verteporfin | Sigma | SML0534 | |
| Software, algorithm | MINS | *Lou et al., 2014* | | |
| Software, algorithm | GraphPad Prism | www.graphpad.com | RRID:SCR_015807 | |
| Software, algorithm | Fiji | fiji.sc | RRID:SCR_002285 | |
| Software, algorithm | R Project for Statistical Computing | www.r-project.org | RRID:SCR_001905 | |

## Animal experimentation

The following mouse lines were used in this work: *CBF1-VENUS* (*Nowotschin et al., 2013*), *Rbpj* null (*Oka et al., 1995*), *Tead4* null (*Nishioka et al., 2008*), *Notch1* null (*Conlon et al., 1995*), *iChr-Notch-Mosaic* (*Pontes-Quero et al., 2017*), *iChr-Control-Mosaic* (*Pontes-Quero et al., 2017*), *Polr2a$^{CreERT2}$* (*Guerra et al., 2003*). All the lines were maintained in heterozygosis in an outbred background. Adults were genotyped by PCR of tail-tip DNA using primers and conditions previously described for each line. For preimplantation embryos, genotyping was performed directly on individually isolated embryos after recovery, culture or antibody staining.

Mice were housed and maintained in the animal facility at the Centro Nacional de Investigaciones Cardiovasculares (Madrid, Spain) in accordance with national and European Legislation. Procedures were approved by the CNIC Animal Welfare Ethics Committee and by the Area of Animal Protection of the Regional Government of Madrid (ref. PROEX 196/14).

## Embryo collection and culture

Females from the different mouse lines or outbred CD1 were superovulated as previously described (*Behringer et al., 2014*), except in the case of embryos to be used for RNA-seq. For embryo culture, zygotes were collected from oviducts, treated with hyaluronidase (Sigma) to remove cumulus cells and cultured until the desired stage at 37.5°C, 5% $CO_2$, in M16 medium (Sigma) covered with mineral oil (NidOil, EMB). For experiments that did not require culture, embryos were collected at morula or blastocyst stage by flushing the oviduct or the uterus with M2 medium (Sigma) and fixed.

## Immunofluorescence of preimplantation embryos

Immunofluorescence was performed as previously described (*Dietrich and Hiiragi, 2007*). The following antibodies and dilutions were used: monoclonal mouse anti-CDX2 (MU392-UC, BioGenex) 1:200, rabbit monoclonal anti-CDX2 (ab76541, Abcam) 1:200, mouse monoclonal anti-YAP (sc-101199, Santa Cruz Biotechnology) 1:200, rabbit polyclonal anti-pERM (3141, Cell Signalling) 1:250, rat monoclonal anti-E-Cadherin (U3254, Sigma) 1:250, mouse monoclonal anti-TEAD4 (ab58310, Abcam) 1:100, rabbit polyclonal anti-DsRed (632496 living colors Clontech) 1:400, goat polyclonal anti-GFP (R1091P, Acris, Origene) 1:200, rat monoclonal anti-HA (11867423001, Sigma) 1:200. Secondary Alexa Fluor conjugated antibodies (Life Technologies) were used at 1:1000. Nuclei were visualized by incubating embryos in DAPI at 1 µg/ml.

## Imaging and quantification

Images of antibody-stained embryos were acquired on glass-bottomed dishes (Ibidi or MatTek) with a Leica SP5, Leica SP8 or Zeiss LSM880 laser scanning confocal microscopes. The same parameters

were used for imaging each experiment. Semi-automated 3D nuclear segmentation for quantification of fluorescence intensity was carried out using MINS, a MATLAB-based algorithm (http://katlab-tools.org/) (*Lou et al., 2014*), and analysed as previously described (*Saiz et al., 2016*). To correct z-associated attenuation, intensity levels were fit to a linear model. Mitotic and pyknotic nuclei were excluded from the analysis. For defining cells as positive or negative for a given nuclear marker, we ordered cells by intensity levels and established a threshold for each experiment based on manual verification of the point where nuclear and cytoplasmic signals were equal. This process was repeated independently for each set of embryos processed and imaged in parallel, to overcome inter-experimental variability.

For live imaging, embryos were cultured in microdrops of KSOM on glass-bottomed dishes (Mat-Tek) in an environmental chamber as described previously (*Xenopoulos et al., 2015*). Images were acquired with a Zeiss LSM880 laser scanning confocal microscope system using a 40x objective. An optical section interval of 1.5 µm was acquired per z-stack, every 15 min.

Cell tracking of 3D-movies was carried out using a TrackMate plugin in Fiji (*Fernández-de-Manúel et al., 2017*; *Schindelin et al., 2012*; *Tinevez et al., 2017*). The 3D reconstruction of the embryos and position of the cells was done using MatLab. The shape of the embryos was fitted into an ellipse and the coordinates in X, Y, Z for each blastomere were normalised to the centroid of the ellipse. The algorithm assigned an inner or outer position to each blastomere according to an established threshold, and they were manually verified. The intensity levels of VENUS fluorescent protein in each cell and time point were normalised according to the Z-position to correct the decay of signal intensity due to the distance with the objective (*Saiz et al., 2016*). The frequencies of the intensity levels for each embryo followed a Gaussian distribution. In order to compare different embryos, intensity levels were normalised so that the mean was 0 and the standard deviation was 1.

## Pharmacological inhibitor treatments

Two-cell or morula stage embryos were cultured in drops of M16 medium (Sigma) covered with mineral oil (NidOil, EMB) at 37°C, 5% $CO_2$, containing the corresponding pharmacological inhibitor or only DMSO as control until the corresponding stage. The following inhibitors and concentrations were used: 10 or 20 µM of the γ-secretase inhibitor RO4929097 (S1575, Selleckchem) (*Münch et al., 2013*) and 10 µM of the TEAD/YAP inhibitor Verteporfin (SML0534, Sigma) (*Liu-Chittenden et al., 2012*).

## Quantitative-PCR

RNA from pools of 25–30 embryos (for pharmacological inhibitor experiments) or from single embryos (for CRISPR/Cas9 editing) was isolated using the Arcturus PicoPure RNA Isolation Kit (Applied Biosystems) and reverse transcribed using the Quantitect Kit (Qiagen). RNA was isolated from ES cells with the RNeasy Mini Kit (Qiagen) and reverse transcribed using the High Capacity cDNA Reverse Transcription Kit (Applied Biosystems). cDNA was used for quantitative-PCR (qPCR) with Power SYBR Green (Applied Biosystems) in a 7900HT Fast Real-Time PCR System (Applied Biosystems). Expression of each gene was normalized to the expression of the housekeeping genes *Actin* (in mESC or pools of embryos) or *18S rRNA* (in single embryos). Primers used are detailed in *Supplementary file 1*.

## RNA-sequencing data analysis

RNA-seq was performed on single morulae. cDNA synthesis was performed using SMART-Seq Ultra Low Input RNA Kit (Clontech). Library preparation and sequencing was performed by the CNIC Genomics Unit using the Illumina HiSeq 2500 sequencer. Gene expression analysis was performed by the CNIC Bioinformatics Unit. Reads were mapped against the mouse transcriptome (GRCm38 assembly, Ensembl release 76) and quantified using RSEM v1.2.20 (*Li and Dewey, 2011*). Raw expression counts were then processed with an analysis pipeline that used Bioconductor packages EdgeR (*Robinson et al., 2010*) for normalisation (using TMM method) and differential expression testing. Expression data of *Rbpj* and *Neo* were used to genotype the samples. Three mutant and three control (two wildtype and one heterozygote) embryos were selected for analysis. Changes in gene expression were considered significant if associated to Benjamini and Hochberg adjusted p-value<0.05.

RBPJ binding motifs were located according to the consensus motif from CIS-BP database (M6499_1.02 motif) using FIMO (*Grant et al., 2011*). Association of RBPJ motifs to DEG was performed using BEDTOOLS (*Quinlan and Hall, 2010*) using a 10 Kb window surrounding the transcriptional start site of genes. ATAC-seq data from 8-cell stage embryos (*Wu et al., 2016*) was mapped to the GRCm38 assembly and integrated with the coordinates of RBPJ motifs previously detailed.

Sequencing data have been deposited at GEO under accession number GSE121979.

## Transient transgenic assay

For the generation of transient transgenics, F1 (C57Bl/6 x CBA) females were superovulated to obtain fertilized oocytes as described (*Behringer et al., 2014*). Each construct was microinjected into the pronucleus of fertilized oocytes at E0.5 at a concentration of 2 ng/μl. Microinjected oocytes were cultured in microdrops of M16 medium (Sigma) covered with mineral oil (NidOil, EMB) at 37°C, 5% $CO_2$ until the morula stage.

Each fragment to be tested was amplified from mouse genomic DNA using NEBuilder HiFi DNA Assembly kit (New England Biolabs) and cloned into a modified pBluescript vector (*Yee and Rigby, 1993*) containing a H2BmRFP reporter gene under the control of the human beta-globin minimal promoter and including an SV40 polyadenylation signal. Primers for amplifying and cloning the 700 bp Tle4-up region are ctatagggcgaattggagctcTTCTTTAGAGGCACCAGTC and ggatccactagttctagagcggccgcATAAAGCCATTTTGCTTAACTG. Primers to amplify and clone the 600 bp Tbx3-i7 region are ctatagggcgaattggagctcCAAGCCAGCCTCAGTCCC and ggatccactagttctagagcggccgcCACACAAGCTTGCCAGCC. Lower case indicates sequence annealing to the plasmid and capital letters indicates sequence annealing to the genome. Constructs were linearized and plasmid sequences removed before microinjection. For H2BmRFP detection, embryos were fixed in 4% paraformaldehyde for 10 min at room temperature and immunostained. Any embryo showing at least one cell expressing the reporter was scored as positive. Due to mosaicism and variability in the amount of transgene present per cell, signal intensity of the reporter cannot be used as a reliable measure of enhancer activity in these assays. When using an empty vector containing only the minimal promoter and the reporter as a negative control, we routinely obtained H2BmRFP expression in approximately 10% of embryos (*Rayon et al., 2014*).

## Mutagenesis

Mutated version of Tle4-up (Tle4-up mut[RBPJ]) was generated by site-directed mutagenesis (Mutagenex Inc), changing the TGTGGGAAA binding motif to TGTccGAAA. Mutated version of Tbx3-i7 (Tbx3-i7 mut[RBPJ]) was generated using QuickChange II XL Site-Directed Mutagenesis Kit (Agilent Technologies) changing CGTGGGAAA to CGTccGAAA. Lower case indicates the altered residues. Changes that abolish RBPJ binding were based on previously described mutated versions of the binding site (*Tun et al., 1994*).

## CRISPR/Cas9 genome editing

Two guide-RNAs at 60 ng/μl were incubated with tracRNA (Sigma) at 240 ng/μl for 5 min at 95°C. The hybridised gRNAs were then incubated with the Cas9 protein (PNA bio) at 30 ng/μl for 15 min at RT and microinjected into the pronuclei of (CBAxC57) F1 zygotes. sgRNAs were designed using the CRISPOR tool (http://crispor.tefor.net/) (*Haeussler et al., 2016*). The following guide RNAs were used: *Tle4*, TTAGCCTGCACTTCGAGTTA and CCCAATTCAAGGCGTTCTGT; *Tbx3*, TAACCCTTTAGAGATAGGCT and TACCAGAGAGGTTTCCTACT. Embryos were recovered at E2.5 and lysed in 50 μl extraction buffer from the Arcturus PicoPure RNA Isolation Kit (Applied Biosystems). Aliquots of 10 μl were used for DNA extraction for PCR genotyping. Mosaic embryos were those where we detected both the deleted and the wildtype allele. The remaining 40 μl were used for RNA extraction for RT-qPCR. To characterize the deletions generated, after PCR genotyping deleted bands of some embryos were gel purified, cloned and sequenced.

## Cell culture

iChr-Notch-Mosaic ES cells were generated by Rui Benedito at CNIC and have been tested negative for *Mycoplasma* by the CNIC Cell Culture Facility. Cells were cultured in standard ESC media (DMEM, Gibco) supplemented with 15% foetal bovine serum (HyClone), 1% Glutamine, 1% NEAA

(Hyclone), 0.1% ß-mercaptoethanol (Sigma) and LIF (produced in-house) or 2i (CHIR-99021, Selleck-chem; and PD0325901, Axon) in dishes seeded with a feeder layer of mouse embryonic fibroblasts (MEFs). Cells were transfected with a *Cre* expressing plasmid to induce recombination using Lipofectamine 2000 (Invitrogen) for 24 hr. After recombination, cells were sorted using a Becton Dickinson FACS Aria Cell Sorter. To promote spontaneous differentiation, cells were cultured on gelatine-covered dishes for 48–72 hr after LIF or 2i + LIF removal in DMEM (Gibco) supplemented with 20% serum, 1% Glutamine and 0.1% ß-mercaptoethanol (Sigma).

## Statistics

Statistical analyses were performed with GraphPad Prism seven or R studio. Data are presented as means ± s.d. or means ± s.e.m. as indicted in each case. Differences were considered statistically significant at p-value<0.05. Tests used to calculate p-value are detailed in the figure legends. Student's t-test was used to compare two groups. ANOVA with Fisher or Bonferroni post-test was used to compare several groups. Fisher's exact test was used to compare distributions.

## Acknowledgements

We thank Alba Álvarez for help with bioinformatics; Laura Fernandez de Manuel and Daniel Jimenez-Carretero for help with cell tracking; Jose Luis de la Pompa, Hiroshi Sasaki and Miguel Torres for mice lines; Manuel J Gomez from the CNIC Bioinformatics Unit, the CNIC Genomics and Microscopy Units for technical support; and members of Manzanares lab for suggestions and discussions. This work was supported by the Spanish government (grants BFU2017-84914-P and BFU2015-72319-EXP to MM; FPI-SO Fellowship to SM); and grants NIH-R01DK084391, NIH-R01HD094868 and NIH-P30CA008748 to AKH. The CNIC is supported by the Spanish Ministry of Science, Innovation and Universities and the Pro CNIC Foundation, and is a Severo Ochoa Center of Excellence (SEV-2015–0505).

## Additional information

### Funding

| Funder | Grant reference number | Author |
|---|---|---|
| Ministerio de Economía y Competitividad | BFU2017-84914-P | Sergio Menchero<br>Isabel Rollan<br>Antonio Lopez-Izquierdo<br>Maria Jose Andreu<br>Julio Sainz de Aja<br>Javier Adan<br>Teresa Rayon<br>Miguel Manzanares |
| ProCNIC Foundation | | Sergio Menchero<br>Isabel Rollan<br>Antonio Lopez-Izquierdo<br>Maria Jose Andreu<br>Julio Sainz de Aja<br>Javier Adan<br>Rui Benedito<br>Teresa Rayon<br>Miguel Manzanares |
| National Institutes of Health | NIH-R01DK084391 | Minjung Kang<br>Anna-Katerina Hadjantonakis |
| Ministerio de Economía y Competitividad | BFU2015-72319-EXP | Sergio Menchero<br>Isabel Rollan<br>Maria Jose Andreu<br>Miguel Manzanares |

| Ministerio de Economía y Competitividad | SEV-2015-0505 | Sergio Menchero<br>Isabel Rollan<br>Antonio Lopez-Izquierdo<br>Maria Jose Andreu<br>Julio Sainz de Aja<br>Javier Adan<br>Rui Benedito<br>Teresa Rayon<br>Miguel Manzanares |
|---|---|---|
| Ministerio de Economía y Competitividad | SVP-2013-067930 | Sergio Menchero |
| National Institutes of Health | NIH-R01HD094868 | Minjung Kang<br>Anna-Katerina Hadjantonakis |
| National Institutes of Health | NIH-P30CA008748 | Minjung Kang<br>Anna-Katerina Hadjantonakis |

The funders had no role in study design, data collection and interpretation, or the decision to submit the work for publication.

### Author contributions

Sergio Menchero, Conceptualization, Data curation, Software, Formal analysis, Validation, Investigation, Visualization, Methodology, Writing—original draft, Project administration, Writing—review and editing; Isabel Rollan, Investigation, Methodology, Writing—review and editing; Antonio Lopez-Izquierdo, Data curation, Software, Formal analysis, Investigation, Visualization, Methodology, Writing—review and editing; Maria Jose Andreu, Javier Adan, Investigation, Writing—review and editing; Julio Sainz de Aja, Formal analysis, Investigation, Writing—review and editing; Minjung Kang, Data curation, Investigation, Writing—review and editing; Rui Benedito, Resources, Writing—review and editing; Teresa Rayon, Conceptualization, Writing—review and editing; Anna-Katerina Hadjantonakis, Conceptualization, Resources, Funding acquisition, Writing—review and editing; Miguel Manzanares, Conceptualization, Supervision, Funding acquisition, Validation, Writing—original draft, Project administration, Writing—review and editing

### Author ORCIDs

Sergio Menchero http://orcid.org/0000-0003-4592-7259
Anna-Katerina Hadjantonakis http://orcid.org/0000-0002-7580-5124
Miguel Manzanares https://orcid.org/0000-0003-4849-2836

### Ethics

Animal experimentation: This study was performed in strict accordance with national and European Legislation. Procedures were approved by the CNIC Animal Welfare Ethics Committee and by the Area of Animal Protection of the Regional Government of Madrid (ref. PROEX 196/14).

### Decision letter and Author response

Decision letter https://doi.org/10.7554/eLife.42930.039
Author response https://doi.org/10.7554/eLife.42930.040

## Additional files

### Supplementary files

• Supplementary file 1. Primers used in this study.
DOI: https://doi.org/10.7554/eLife.42930.030

• Transparent reporting form
DOI: https://doi.org/10.7554/eLife.42930.031

### Data availability

Sequencing data have been deposited in GEO under accession code GSE121979.

The following dataset was generated:

| Author(s) | Year | Dataset title | Dataset URL | Database and Identifier |
|-----------|------|---------------|-------------|-------------------------|
| Menchero S | 2018 | Transitions in cell potency during early mouse development are driven by Notch | https://www.ncbi.nlm.nih.gov/geo/query/acc.cgi?acc=GSE121979 | NCBI Gene Expression Omnibus, GSE121979 |

The following previously published datasets were used:

| Author(s) | Year | Dataset title | Dataset URL | Database and Identifier |
|-----------|------|---------------|-------------|-------------------------|
| Mubeen Goolam, Antonio Scialdone, Sarah J L Graham, Iain C Macaulay, Agnieszka Jedrusik, Anna Hupalowska, Thierry Voet | 2016 | Single-cell RNA-seq of blastomeres from 2- to 32-cell stage mouse embryos | https://www.ebi.ac.uk/arrayexpress/experiments/E-MTAB-3321/ | Array Express, E-MTAB-3321 |
| Xie W | 2016 | The landscape of accessible chromatin in mammalian pre-implantation embryos | https://www.ncbi.nlm.nih.gov/geo/query/acc.cgi?acc=GSE66390 | NCBI Gene Expression Omnibus, GSE66390 |

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
