## [Decision Letter]

Thank you for submitting your article "Transitions in cell potency during early mouse development are driven by Notch" for consideration by *eLife*. Your article has been reviewed by three peer reviewers, and the evaluation has been overseen by a Reviewing Editor and Kevin Struhl as the Senior Editor. The reviewers have opted to remain anonymous.

The reviewers have discussed the reviews with one another and the Reviewing Editor has drafted this decision to help you prepare a revised submission.

Summary:

All three reviewers were enthusiastic about the findings suggesting a role for Notch signalling in regulating *Cdx2* expression in the pre-implantation mouse embryo, and the identification of *Tle4* as a Notch-signalling target and agree that the work is, in principle, very suitable for publication in *eLife*. You will see that the reviewers require substantial additional experimental support for your conclusions.

Essential revisions:

In revising the manuscript we would ask you in particular to address the four main concerns raised during the review process, namely 1) to improve the data to substantiate your claim that Notch signalling regulates cell positioning 2) to increase the number of *Rbpj^-/-^* embryos subjected to single cell sequencing 3) strengthening of the data that shows that Notch and Hippo operate in parallel and 4) a more in-depth analysis of the differentiation potential of iChr-Notch mosaic ES cell.

The issues raised with regard to these points are clearly articulated in the reviewers’ comments below.

*Reviewer #1:*

During the preimplantation period of mammalian development, cell potency is gradually diminished as cells adopt distinct fates, including inner cell mass and trophectoderm. This interesting study from Menchero and colleagues examines the role of the Notch signaling pathway in regulating expression of *Cdx2* in mouse preimplantation embryos. CDX2 is widely acknowledged to be an important regulator of trophectoderm cell fate, and the role of Hippo signaling in regulating *Cdx2* is well established. By comparison, the role of Notch signaling in regulating CDX2 and cell fate is a relatively new topic of inquiry, as is the connection between Notch and Hippo signaling.

The authors employ a variety of clever and sophisticated new genetic tools to manipulate and measure Notch signaling in the embryo. From their analyses, the authors conclude that Notch signaling promotes expression of the trophectoderm gene *Cdx2* starting at the 4-cell stage. They conclude that Notch signaling is active before the stage that Hippo signaling is active in embryos. They further claim that Notch signaling dictates cell position, which is important because cell position is thought to dictate trophectoderm versus inner cell mass fate. They also identify new targets of Notch signaling by RNA-seq and evaluate the roles of these genes in ES cell differentiation.

All of these findings would be of broad interest to embryologists, stem cell biologists, and signaling pathway enthusiasts. However, the evidence to support the study's most exciting conclusions are still quite preliminary (see below). Therefore, while this study has the potential to provide exciting new insight to signaling in blastocyst formation, the story does not yet rest on a concrete foundation, and it's really hard to know if the authors' model is accurate.

• Replication unknown. For the majority of figures, the sample size is not reported (neither for number of embryos examined nor for the number of embryo cells examined). Therefore, it is not possible to evaluate whether the reported phenotypes are highly penetrant or even representative.

• Ambiguous embryo staging. Throughout the manuscript, the authors refer to embryos at the morula stage. However, the morula stage encompasses several dynamic stages (8-cell to <32-cell stage). Therefore, it is not possible to know that the comparisons are apples-to-apples across different litter, treatments, and culture conditions, without a more granular view (precise embryo cell number counts). For example, in Figure 1—figure supplement 4, are the *Tead4* null embryos the same cell number? Could this explain the increase in CDX2 expression (as opposed to overexpression of NICD)

• In several places, the data seem overinterpreted. For example:

- Figure 2: the claim is made that Verteporfin and RO4929097 inhibit YAP/TEAD and Notch signaling, respectively. However, the authors do not provide an independent assessment of the efficacy of these particular treatments. Does each treatment window allow RO inhibit expression of the N signaling reporter CBF1? Does Verteporfin inhibit the nuclear localization of YAP (as shown by Wang et al., 2016)? While the authors provided some of these data in their previous manuscript (Rayon et al., 2014), the treatment windows differ between the two studies. Without knowledge of the degree of inhibition, it is not possible to rule out the possibility that a signaling pathway is active in a particular time or place.

- Figure 4: the authors state that N signaling determines cell positioning. However, the authors have not confirmed whether the genetic manipulations (overexpression of NICD or dominant-negative MAML indeed affect N signaling in this setting as assumed. Another concern with this figure is whether an increase in ICM contribution is meaningful, when there were so few N loss of function cells surveyed to begin with (n = 6 cells). Increasing the sample size would be one way to address this. Finally, it is nice that wild type, unrecombined cells serve as an internal control in these experiments. However, I am slightly concerned that there is no control to examine whether the fluorescent proteins themselves or some other feature not related to Notch, could skew cellular contributions. This is a relatively minor concern. Finally, "Different N levels determines cell position" seems an overly strong conclusion, since many cells with altered N signaling do not change position.

– For the RNA-seq analysis of *Rbpj* null embryos, only two null embryos were evaluated (Figure 5). This probably does not provide adequate biological replication and could explain why one of the proposed RBPJ targets (*Tbx3*) was a false positive.

• In some places, the data do not support the conclusions stated.

- The conclusion that Notch, but not YAP/TEAD4 are essential for regulating expression of *Cdx2* prior to blastocyst formation is based on analysis of qPCR data shown in Figure 2A, where *Cdx2* levels are compared between treated and control embryos (left and right panels). However, if you compare the *Cdx2* expression levels in RO-treated and Verteporfin-treated embryos, their levels are similar. By contrast, it is the variation within the control embryos that makes it seem like the effect of RO is statistically significant while the effect of the Verteporfin is not. Finally, do Rbpj null embryos phenocopy RO treatment at the early/morula stage?

- Subsection “Absence of crosstalk between the Notch and Hippo signalling pathways in the early mouse embryo”, last paragraph: if the maternally provided Cre recombines in oocytes, then no mosaicism should be observed in embryo cell recombination. Do the authors mean that the maternally provided Cre recombines during the embryonic stages? Probably this is just a semantic issue.

*Reviewer #2:*

Having previously shown that Notch signalling regulates the expression of CDX2 in conjunction with YAP/TEAD complexes in TE cells, the authors investigate the role of the pathway during early preimplantation development. Using a variety of approaches (pharmacological, genetic), the authors claim that Notch plays an unexpected role in regulating genes that act as repressors of early naïve pluripotency prior to compaction, thereby initiating differentiation events that drive early specification of the trophectoderm.

The manuscript is clearly written and presents potentially interesting findings. However, in some cases, it was let down by either its experimental design, a lack of extensive controls or an inappropriate number of independent biological replicates. Please find below a list of major points that I think need to be addressed before the manuscript is ready for publication

1) The importance of Notch signalling during early preimplantation development

If Notch signalling is crucial prior to compaction to initiate TE differentiation, it is not clear why *Rbpj^-/-^* embryos develop into apparently morphologically normal blastocysts (Figure 1—figure supplement 2) and have also completely recovered a normal CDX2 expression pattern after the 16 cell-stage (Figure 2E-F).

2) Lack of quantification of CDX2 nuclear levels in mutant embryos

The authors present results obtained with *Rbpj^-/-^* embryos and the impact on CDX2 expression in a separate figure to Figure 1 which makes comparisons between the various genotypes difficult. CDX2 nuclear levels seem lower in *Rbpj^-/-^; Tead4^+/-^* and *Rbpj^+/-^; Tead4^-/-^* embryos (Figure 1A), whereas in *Rbpj^-/-^* embryos cells are either described as positive or negative for CDX2 (Figure 2C). This discrepancy is unexplained in the manuscript but proper quantification of nuclear CDX2 would go some way in properly describing the results and allow comparisons between embryos of different genotypes.

3) Absence of description of *Tead4^-/-^* embryos

It seems that the data for *Tead4^-/-^* embryos is not presented in Figure 1A and elsewhere in the manuscript. Since a previous report suggests that TEAD4 accounts for most of CDX2 expression in the nucleus at E2.5 (Yagi et al., 2007), it needs to be addressed whether it is truly the case to find what is the exact contribution of both Notch and YAP/TEAD4. In particular at the 8-cell stage, since the authors claim that "there is an earlier requirement for Notch than for Hippo in the regulation of *Cdx2*, and that both pathways exert non-redundant roles".

4) Weak effect of N1ICD in *Tead4^-/-^* embryos on CDX2 expression

In Figure 1—figure supplement 4, expressing N1ICD in *Tead4^-/-^* embryos seems to have a marginal effect at best (albeit significant) as nuclear CDX2 intensity levels go up from around 2.25 to 2.50. It seems that this was quantified at the E3.5. Considering the authors' conclusions regarding the role of Notch signalling prior to the morula stage, it seems that it would be more relevant to investigate the effect of N1ICD in *Tead4^-/-^* embryos at earlier stages and it may be that nuclear CDX2 expression is rescued more impressively at these stages. Additionally, to further asses the relative importance of Notch and *Tead4* at early stages, the reverse experiment would be to try and rescue nuclear CDX2 expression in *Rbpj^-/-^* embryos using a constitutively active form of TEAD4 for instance.

5) Lack of clarity in the GOF and LOF regarding recombination events

In the GOF and LOF experiments, the authors resort to immunostaining to identify the different cell recombination events. However, the GOF cells were triple positive because of antibody cross-reactivity. In addition, the images presented are unclear and, the examples given, fail to alleviate concerns about the validity of the approach and why it was used. If fixation reduced the fluorescence signal, could the embryos have been imaged live prior to that? Alternatively, have the authors attempted to validate their method by other means (by genotyping after immunostaining of individual dissociated blastomeres for example)? The fact that the less likely recombination event was observed the most often using this method adds to the concern that it may not accurately identify the right type of cell.

6) Number of independent biological replicates in RNA-seq experiments

Only 2 *Rbpj^-/-^* morulae were used in RNA-seq experiments. It seems unreasonable to justify this by the decision of wanting to strictly compare litter mates.

7) Quantification of transient transgenic experiment

Figure 5E-F – It is not clear why the authors quantified positive and negative embryos. Unless I misunderstood the experiment, it seems that embryos injected with the mutated transcriptional enhancer should either drive H2B-mRFP weakly or not at all in comparison to the wild type construct. As a result, it seems that signal intensity should have been quantified instead.

8) CRISPR experiment

Figure 5G – Not enough information is provided to understand what mutations were obtained in the CRISPR experiment. Was a repair construct lacking the enhancer sequence introduced to generate a knockin in the endogenous locus? If not, were the different mutated embryos harbouring the same mutations?

*Reviewer #3:*

The current manuscript by Menchero and colleagues describes an analysis of the role of Notch signalling during early pre-implantation development. The authors had previously shown that Notch and Yap/Tead signalling act together to specify trophectoderm (TE) identity. Here using a series of elegant single cell expression analyses combined with genetic and inhibitor experiments they demonstrate that these are parallel pathways, with Notch signalling acting earlier than Hippo signalling to affect *Cdx2* expression and control TE fate. Importantly, by studying null mutants for the Notch effector *Rbpj* as well as using mosaic Notch gain and loss of function mutations, they also identify that Notch signalling appears to regulate not only TE fate, but have a more general role in promoting differentiation.

Together, these are very interesting findings that provide important new insight into pre-implantation development. The role of FGF and Hippo signalling have been extensively analysed, therefore insight into the roles of other pathways is welcome. The analysis of the relative roles of Notch and Hippo on the regulation of TE fate is conclusive. In contrast to this, the manuscript would benefit from some clarification regarding the role of Notch signalling in the exit of naive pluripotency. Specifically, it would be important to understand if *Rbpj*/Notch signalling is required for this process or if the regulation of *Prdm14* and *Dppa3* is reflecting other roles of Notch signalling. Therefore, it would be interesting to have a broader picture of the role of Notch in exit of pluripotency and if *Tle4* is an important target during this process. Using naïve pluripotent ES cells (2i+Lif) and their differentiation will most likely be the most straightforward way to address this question.

---

## [Author Response]

Essential revisions:In revising the manuscript we would ask you in particular to address the four main concerns raised during the review process, namely:1) To improve the data to substantiate your claim that Notch signalling regulates cell positioning.

We have further substantiated our claim that Notch signalling regulates cell position in the preimplantation embryo and the validity of the Notch mosaic mouse model by:

- increasing the number of cells examined at the morula stage in the iChr-Notch-Mosaic model.

- using non-recombined blastomeres in morula and blastocyst mosaic as a second internal control.

- checking the specificity of the antibody panel used to distinguish different recombination events in embryos by testing them in fluorescent-based sortied ES cells pf the same genotype as the iChr-Notch-Mosaic mice.

- including a new control using a mouse line carrying the same construct as the iChrNotch-Mosaic, but that does only carry the fluorescent proteins with no fusions to DNMaml1 or NICD (iChr-Control-Mosaic).

These results are now shown in revised Figure 4, Figure 4—figure supplement 1, and Figure 4—figure supplement 3.

2) To increase the number of Rbpj^-/-^ embryos subjected to single cell sequencing.

We have included a further *Rbpj*^-/-^ embryo in the RNA-seq and re-done the analysis, confirming our previous results. These results are now shown in revised Figure 5, Figure 5—figure supplement 1, Figure 5—source data 1, and Figure 5—source data 2.

3) Strengthening of the data that shows that Notch and Hippo operate in parallel.

We have considerably strengthened the data that shows that Notch and Hippo act in parallel by:

- separating the data of the correlation analysis of CDX2, Yap and VENUS for morulas in two different windows based on cell number (8-16 and 17-32 cells).

- increasing the number of Verteporfin treated embryos analysed by RT-qPCR in different time windows.

- including new controls that show that drug treatments work as expected in these time windows by decreasing fluorescence from the CBF:VENUS reporter or downregulating known YAP-TEAD4 targets in preimplantation embryos.

These results are now shown in revised Figure 1, Figure 1—figure supplement 2, Figure 2, and Figure 2—figure supplement 1.

4) A more in-depth analysis of the differentiation potential of iChr-Notch mosaic ES cell.

We have carried out novel differentiation experiments of ES cells with different Notch levels that were grown and maintained in 2i+LIF conditions, in order to examine the their potential starting from the naïve state. These results are now shown in revised Figure 6—figure supplement 4.

The issues raised with regard to these points are clearly articulated in the reviewers’ comments below.

See response to each of the issues raised by the reviewers below, along with the description of the corresponding changes in the main text and in the figures.

Reviewer #1:[…] All of these findings would be of broad interest to embryologists, stem cell biologists, and signaling pathway enthusiasts. However, the evidence to support the study's most exciting conclusions are still quite preliminary (see below). Therefore, while this study has the potential to provide exciting new insight to signaling in blastocyst formation, the story does not yet rest on a concrete foundation, and it's really hard to know if the authors' model is accurate.• Replication unknown. For the majority of figures, the sample size is not reported (neither for number of embryos examined nor for the number of embryo cells examined). Therefore, it is not possible to evaluate whether the reported phenotypes are highly penetrant or even representative.

We are sorry for not including this basic information and thanks the reviewer for pointing it out. We have carefully revised all of the data presented in the manuscript and included sample size where needed. For example, in the case of CDX2 staining in the different allelic combinations of the *Rbpj* and *Tead4* mutants, we have included the number of embryos analysed on top of the corresponding panels (Figure 1A and Figure 1—figure supplement 1A). In other cases (Figure 1—figure supplement 3, Figure 1—figure supplement 4, Figure 2—figure supplement 1) we have included this information in the figure legends. In all other cases, samples sizes (number of embryos and/or cells) are indicated in the figure legends.

• Ambiguous embryo staging. Throughout the manuscript, the authors refer to embryos at the morula stage. However, the morula stage encompasses several dynamic stages (8-cell to <32-cell stage). Therefore, it is not possible to know that the comparisons are apples-to-apples across different litter, treatments, and culture conditions, without a more granular view (precise embryo cell number counts). For example, in Figure 1—figure supplement 4, are the Tead4 null embryos the same cell number? Could this explain the increase in CDX2 expression (as opposed to overexpression of NICD)

We agree with the reviewer that cell number counts is a more accurate description of embryos stages in these time windows, and where possible we have separated the data from embryos in early (8-16 cells) and late (17-32) stages. We have done so for the correlations analysis of levels of CDX2, YAP and VENUS shown in Figure 1C and Figure 1—figure supplement 2, allowing us to observe nicely the gradual increase in correlation along development of CDX2 with VENUS (as a readout of Notch activity) and YAP, while correlation of Venus with YAP remains low until the blastocyst. We also show this distinction when we quantify nuclear levels of CDX2 in *Rbpj* and *Notch1* mutant embryos (Figure 2C-H and Figure 2—figure supplement 2A-F). We have also changed the image of the double heterozygous *Rbpj^+/-^;Tead4^+/-^* shown in Figure 1A so it is at the same 8-cell stage as other embryos shown in this panel (see response to reviewer #2 below).

In the case of the pharmacological treatments (Figure 2A, B and Figure 5C), we could not accurately count cell numbers per embryo as these where live and directly placed in lysis buffer for RNA extraction. Manipulation such as fixing and staining nuclei would have compromised the quality of the RNA, thus not allowing to perform RT-qPCR on this limited amount of material.

As for the experiment of the rescue of the *Tead4* mutant phenotype by overexpression of N1ICD (shown in Figure 1—figure supplement 4 of the original version of the manuscript), we followed the reviewer’s advice and carefully counted the number of cells of the embryos analysed. While on average embryos had 40-65 cells, one of the *Tead4^/-^* embryos was considerably earlier with only 25 cells. We therefore excluded this embryo from the analysis and included two new *Tead4^-/-^* embryos with a cell number within the range of the others previously analysed. When we repeated the comparison of nuclear CDX2 intensity levels between *Tead4^-/-^* and *N1ICD-GFP;Tead4^-/-^* embryos, we found no significant differences (Author response image 1). In light of this result, and taking into account that we had previously argued that N1ICD only marginally rescues CDX2 expression levels in *Tead4^-/-^* but not its developmental phenotype, we have decided to remove this information from the present version of the manuscript (see also response to point 4 from reviewer #2 below).

**Author response image 1. respfig1:** Quantification of CDX2 intensity levels in *Tead4^-/-^* (n=153 blastomeres from 3 embryos) and *N1ICD-GFP;Tead4^-/-^* (n=224 blastomeres from 5 embryos) embryos. No significant differences in CDX2 levels was found between the two genotypes.

• In several places, the data seem overinterpreted. For example:- Figure 2: the claim is made that Verteporfin and RO4929097 inhibit YAP/TEAD and Notch signaling, respectively. However, the authors do not provide an independent assessment of the efficacy of these particular treatments. Does each treatment window allow RO inhibit expression of the N signaling reporter CBF1? Does Verteporfin inhibit the nuclear localization of YAP (as shown by Wang et al., 2016)? While the authors provided some of these data in their previous manuscript (Rayon et al., 2014), the treatment windows differ between the two studies. Without knowledge of the degree of inhibition, it is not possible to rule out the possibility that a signaling pathway is active in a particular time or place.

We agree with the reviewer that we do not formally show that drug treatments in the time windows used result in inhibition of the signalling pathways analysed. We have carried out new experiments to address this issue, treating embryos from the CBF1-VENUS line with the Notch pathway inhibitor RO4929097 from 2-cell to morula and from morula to blastocysts. In both cases, we observe a robust downregulation of VENUS expression, proving that we are effectively inhibition Notch signalling as expected. These results are shown in the new Figure 2—figure supplement 1, and nicely complement our previous findings (Rayon et al., 2014).

As for the effect of Verteporfin on Hippo signalling, it has been described that this drug acts by disrupting the association of the YAP/TEAD complex (Liu-Chittenden et al., 2012). It is true, as the reviewer points out, that the report by Wang et al., 2016 suggests that treatment with Verteporfin causes YAP translocation to the cytoplasm. We therefore examined in detail subcellular localization of YAP in Verteporfin and DMSO (control) treated embryos, but we did not observe changes in nuclear or cytoplasmic localization (Author response image 2).

**Author response image 2. respfig2:** Nuclear localization of YAP is not altered by Verteporfin treatment of preimplantation mouse embryos. *Left*, representative images of DMSO (n=3) and Verteporfin (10 µm; n=4) treated embryos stained for YAP; nuclei, stained with DAPI, are shown below. *Right*, quantification of nuclear YAP levels in DMSO and Verteporfin treated embryos. We observe no difference between treatments.

Nevertheless, we believe this does not mean that the drug is not acting as expected. The above-mentioned paper was carried out in tissue cultured cells, and from the images provided it is difficult to quantitatively assess the extent of nuclear-cytoplasmic shuffling of YAP. An independent way to support our claim that Verteporfin is interfering with YAP/TEAD4 in the time windows used, is to check the expression of known YAP/TEAD4 targets such as *Cdx2* (Yagi et al., 2007; Nishioka et al., 2008; Frum et al., 2018) or *Gata3* (Ralston et al., 2010). Sure enough, we observe a reduction in *Cdx2* expression in Verteporfin treated embryos from morula to blastocyst window (Figure 2B), and of *Gata3* (that is downstream of *Tead4* independently from *Cdx2*; Ralston et al., 2010) in embryos treated from 2-cell to morula (Figure 2A; and see below).

As for the degree of inhibition we obtain, we do not suggest that we are completely blocking the pathways. Pharmacological treatments at these stages are tricky, as high doses results in embryo lethality (usually due to high concentrations of the carrier, in this case DMSO), so at best we obtain a partial reduction. Nevertheless, these experiments also offer unique opportunities to treat the embryos in very precise time windows, what is not possible using genetic approaches.

- Figure 4: the authors state that N signaling determines cell positioning. However, the authors have not confirmed whether the genetic manipulations (overexpression of NICD or dominant-negative MAML indeed affect N signaling in this setting as assumed.

The reviewer points to a relevant issue, although difficult to assess in this experimental setting. An apparently obvious possibility, in view of the tools we have used in our work, would be to use the CBF1:VENUS reporter as a readout of Notch signalling with single cell resolution. However, apart from the difficulties involved in the breeding of three different alleles simultaneously (in terms of space in our animal facility and time), we would not be able to distinguish green fluorescence from the VENUS reporter or from the EGFP fusion to dominant negative MAML1.

The other possibility is to examine the expression of known Notch targets in other contexts, such as *Hes1* or *Hey* genes. However, data from our RNA-seq and that of others show that the levels of expression of these genes in mouse preimplantation development is zero or extremely low. Therefore, we decided to measure nuclear levels of CDX2 as a readout of Notch signalling, as this is the only bona-fide target of Notch identified and validated so far in mouse preimplantation stages. We find a perfect correlation between CDX2 and Notch activity determined by the loss- and gain-of-function systems used. We have included these new results in the revised version of the manuscript and show them in Figure 4—figure supplement 1B.

Another concern with this figure is whether an increase in ICM contribution is meaningful, when there were so few N loss of function cells surveyed to begin with (n = 6 cells). Increasing the sample size would be one way to address this.

We agree with the reviewer this number was somewhat low, so we have setup and analysed more embryos from this cross in order to increase the sample size. We have now analysed 10 morula and 11 blastocysts. We used embryos containing cells derived from all three possible recombination events plus unrecombined cells, and also embryos that at least had recombined cells for both the Notch gain- and loss-of-function events (see response to reviewer #2 below). We have now analysed 29 Notch loss-of-function cells in the morula instead of 6 as before. Alongside, we have also increased the sample size for all other events (unrecombined – see below, wild-type recombined – red, loss-of-function – green, gain-of-function – cyan) at both morula (69, 33, 29 and 83 cells respectively) and blastocyst (190, 87, 50 and 173 cells respectively) stages. Details are provided in the revised legend for Figure 4. In the new analysis, we still observe differential contribution of cells to inner or out positions dependant on Notch activity, although in a more gradual manner. For example, Notch loss-of-function cells are more frequently located in inner positions in the blastocyst (Figure 4E) and also in the morula (Figure 4D; although in this case just below the threshold of statistical significance, p=0.05).

Finally, it is nice that wild type, unrecombined cells serve as an internal control in these experiments.

We thank the reviewer for this nice and very useful suggestion to use unrecombined cells as a second internal control in addition to the Cherry cells. We have counted these cell in both morula and blastocyst stages finding that their distribution in inner or outer positions is identical to recombined Cherry wild type cells (Figure 4D, E). As noted above, in this re-analysis we have included not only embryos in which we observe all three possible recombination events but also those in which we observed at least recombination leading to Notch loss- and gain-of-function cells (see also response to reviewer #2 below). We have updated Figure 4C-E; Figure 4—figure supplement 2A, B; and Figure 4—figure supplement 3B, C (see below) with this data.

However, I am slightly concerned that there is no control to examine whether the fluorescent proteins themselves or some other feature not related to Notch, could skew cellular contributions. This is a relatively minor concern.

The reviewer is right in noting that skews in distribution of cells harbouring different recombination events could be due to factors other than different activation state of the Notch pathway. in order to address this valid and important concern, we have carried out a new set of experiments in which we use a control mice line that carries the same construct as the Notch mosaic, but including only the fluorescent reporters not fused to dominant-negative MAML1 or constitutively active NICD (iChr-Control-Mosaic; PontesQuero et al., 2017), crossed with the *Polr2a^CreERT2^* driver as before (Figure 4—figure supplement 3A). When we examined the contribution of the different recombination events, we find that the majority correspond to Cherry+ cells (Figure 4—figure supplement 3B), the first and most probable event, in contrast to what we observed with the iChrNotch-Mosaic where Cerulean+ Notch gain-of-function cells were more abundant (Figure 4C). This control also shows that the type of recombination event that has taken place does not influence the in-out distribution of cells in the blastocyst (Figure 4—figure supplement 3C). Overall, this new set of control experiments confirms and further supports our previous claim that forced activation of the Notch pathway offer a competitive advantage to Cerulean+ cells that therefore are present in a higher than expected proportion in the resulting embryos.

Finally, "Different N levels determines cell position" seems an overly strong conclusion, since many cells with altered N signaling do not change position.

We do agree with the reviewer that the title of this section was an over-interpretation. We have now changed it to "Different N levels *influences* cell position". We have also changed a sentence in the last section of the Introduction from “differences in Notch levels *determines* cell fate acquisition in the blastocyst” to “differences in Notch levels *contribute to* cell fate acquisition in the blastocyst” along the same line.

- For the RNA-seq analysis of Rbpj null embryos, only two null embryos were evaluated (Figure 5). This probably does not provide adequate biological replication and could explain why one of the proposed RBPJ targets (Tbx3) was a false positive.

We certainly have to agree with the reviewer that n=2 is a low number for analysis of *Rbpj*^-/-^ embryos. In our original analysis this was due to the technical hurdles involved in obtaining adequate material, as embryos where processed in batch and genotypes were only inferred later from sequence data. On the other hand, we have used this approach as an initial discovery tool to identify other putative Notch pathway targets, such as *Tle4*, that we then have confirmed by independent means. Nevertheless, we have carried out new RNA-seq experiments and been able to obtain data from another *Rbpj* null embryo. We have included this data and repeated the analysis, confirming and replicating our previous results and observations. We have updated Figure 5, Figure 5—figure supplement 1 and Figure 5—source data 1 and 2 correspondingly. We have also updated the GEO database entry with the newly generated data.

Regarding *Tbx3*, we still identify it as a differentially expressed gene containing a putative RBPJ site in an open chromatin region in its vicinity. We therefore do not think it is a false positive, as its expression is significantly reduced in RO treated embryos (Figure 5C). The fact that deletion of the putative RBPJ site does not affect enhancer activity in transient transgenic assays, and that the deletion of the corresponding genomic region does not reduce endogenous *Tbx3* expression suggests that this is not the only transcriptional input required. Further unidentified sites could be acting together with this unique RBPJ site, as is the case for the TEE enhancer we previously characterized for *Cdx2* (Rayon et al., 2014). A more detailed analysis of the regulation of *Tbx3* in the preimplantation embryo is of clear interest and will sure merit further work, but we believe it is out of the scope of the present manuscript.

• In some places, the data do not support the conclusions stated.- The conclusion that Notch, but not YAP/TEAD4 are essential for regulating expression of Cdx2 prior to blastocyst formation is based on analysis of qPCR data shown in Figure 2A, where Cdx2 levels are compared between treated and control embryos (left and right panels). However, if you compare the Cdx2 expression levels in RO-treated and Verteporfin-treated embryos, their levels are similar. By contrast, it is the variation within the control embryos that makes it seem like the effect of RO is statistically significant while the effect of the Verteporfin is not.

We have followed the reviewer’s advice and increased the sample size for Verteporfin treated embryos in the 2-cell to morula time window. In this way, we have reduce the variability in DMSO treated embryos and still find no significant differences in *Cdx2* expression (Figure 2A), thus confirming our previous interpretation. Interestingly, what we do observe now is a significant downregulation of *Gata3* upon Verteporfin treatment in this early time window. As discussed above, *Gata3* has been established as a TEAD4 target independent of *Cdx2* (Ralston et al., 2010), so this might be reflecting this early input of Hippo signalling on *Gata3*. As for the differential early and late effects of Notch inhibition on *Cdx2* expression, we have confirmed the RTqPCR results independently by adding the quantification of nuclear CDX2 expression levels in early morula (8-16 cells) from *Rbpj* (Figure 2D, G) and *Notch1* (Figure 2—figure supplement 2B, E) null embryos (see also response to point 2 from reviewer #2 below).

Finally, do Rbpj null embryos phenocopy RO treatment at the early/morula stage?

Yes, as mentioned above we have now included detailed quantification of the nuclear levels of CDX2 in *Rbpj* and *Notch1* null embryos (Figure 2D, G; Figure 2—figure supplement 2B, E), showing a reduction of expression in early 8-16 cell morulas but not in later embryos. This data complements our previous estimate of the number of CDX2 positive cells in these embryos (now shown in Figure 2E, H; Figure 2—figure supplement 2C, F), providing a more fine-grained analysis.

- Subsection “Absence of crosstalk between the Notch and Hippo signalling pathways in the early mouse embryo”, last paragraph: if the maternally provided Cre recombines in oocytes, then no mosaicism should be observed in embryo cell recombination. Do the authors mean that the maternally provided Cre recombines during the embryonic stages? Probably this is just a semantic issue.

The reviewer is right that if the maternal *Sox2-*CRE recombines in the oocyte, we should not expect mosaicism. However, as we noted in our previous work (Rayon et al., 2014), this Cre driver does result in mosaic recombination. Therefore, even if CRE is expressed in the oocyte as previously described (Hayashi et al., 2003, Genesis 37 51-3), it must be acting at later stages. Nevertheless, we have now removed this data from the revised version of the manuscript (see above).

Reviewer #2:[…] The manuscript is clearly written and presents potentially interesting findings. However, in some cases, it was let down by either its experimental design, a lack of extensive controls or an inappropriate number of independent biological replicates. Please find below a list of major points that I think need to be addressed before the manuscript is ready for publication

We thank the reviewer for the comments, which we have addressed in this revised version. Among other new experimental data, we have added controls as requested and also increased the number of biological replicates in key experiments.

1) The importance of Notch signalling during early preimplantation developmentIf Notch signalling is crucial prior to compaction to initiate TE differentiation, it is not clear why Rbpj^-/-^ embryos develop into apparently morphologically normal blastocysts (Figure 1—figure supplement 2) and have also completely recovered a normal CDX2 expression pattern after the 16 cell-stage (Figure 2E-F).

As we discuss extensively in this manuscript, as well as in our previous work (Rayon et al., 2014), we contend that the input of Notch is necessary in these early stages for the proper expression of TE markers such as *Cdx2* and to regulate cell position in the embryo. At no point do we conclude that Notch signalling, or *Rbpj*, is absolutely necessary to specify the TE, but that in acts in concert and along time with other signalling pathways such as Hippo in this process. The lack of an overt morphological phenotype affecting the TE indicates that this pathway is required but that it can be compensated by others triggered independently by cell polarisation. This fits with the hypothesis that the early mammalian embryo acts as a self-organizing system (Wennekamp et al., 2013), integrating genetic, morphological and physical forces to achieve its final form at the blastocyst stage. However subtle, the fact that double *Rbpj^-/-^;Tead4^-/-^*embryos fail to reach the blastocyst stage (Rayon et al., 2014) further supports a role for Notch, together with Hippo, in these early stages.

2) Lack of quantification of CDX2 nuclear levels in mutant embryosThe authors present results obtained with Rbpj^-/-^ embryos and the impact on CDX2 expression in a separate figure to Figure 1 which makes comparisons between the various genotypes difficult. CDX2 nuclear levels seem lower in Rbpj^-/-^; Tead4^+/-^ and Rbpj^+/-^; Tead4^-/-^ embryos (Figure 1A), whereas in Rbpj^-/-^ embryos cells are either described as positive or negative for CDX2 (Figure 2C). This discrepancy is unexplained in the manuscript but proper quantification of nuclear CDX2 would go some way in properly describing the results and allow comparisons between embryos of different genotypes.

We do agree with the reviewer that data at some points is distributed among multiple figures and panels, what can make direct comparisons not easy. We have tried different ways to correct this, but after multiple attempts we still believe that the way the data is presented is the best to follow the story. Regarding quantification of nuclear expression levels shown in Figure 1 for different genotypes, and as discussed the response to reviewer #1 above, we were not able to quantify nuclear expression levels for CDX2 in this set of experiments due to inconsistencies in microscopy settings across different days and litters (see above). However, in an independent set of experiments where we examined nuclear CDX2 expression in morula stage *Rbpj*^-/-^ and *Notch1*^-/-^ embryos, we could carry out correct quantification and compare between embryos. This data is now shown in revised Figure 2D, G and Figure 2—figure supplement 2B, E.

3) Absence of description of Tead4^-/-^ embryosIt seems that the data for Tead4^-/-^ embryos is not presented in Figure 1A and elsewhere in the manuscript. Since a previous report suggests that TEAD4 accounts for most of CDX2 expression in the nucleus at E2.5 (Yagi et al., 2007), it needs to be addressed whether it is truly the case to find what is the exact contribution of both Notch and YAP/TEAD4. In particular at the 8-cell stage, since the authors claim that "there is an earlier requirement for Notch than for Hippo in the regulation of Cdx2, and that both pathways exert non-redundant roles".

We originally did not include this information due to the high number of different genotypes obtained from the cross for the double KO embryos (9 in total), that would have resulted in a too complex figure. However, we did show CDX2 expression in a 8cell stage *Rbpj^+/-^;Tead4^-/-^* embryo that we thought could be representative of the effect of lack of *Tead4* on CDX2 expression. Nevertheless, we have now completed this description by including the immunostaining for CDX2 in all genotypes in Figure 1—figure supplement 1B. Unfortunately, and as described above and in the response to reviewer #1, we could not quantify accurately expression in this experiment, so we can only reach conclusions based on qualitative assessment of antibody stainings. In any case, we do feel sure enough to state that CDX2 is expressed in *Tead4*^-/-^ 8-cell embryos, although at lower levels than what can be observed in wildtypes, and not completely absent as we see in double KO embryos. This is in line with the cited work of Yagi et al. (Yagi et al., 2007) and that of other authors (Nishioka et al., 2008; Frum et al., 2018), that we have included in the revised text of the manuscript.

4) Weak effect of N1ICD in Tead4^-/-^ embryos on CDX2 expressionIn Figure 1—figure supplement 4, expressing N1ICD in Tead4^-/-^ embryos seems to have a marginal effect at best (albeit significant) as nuclear CDX2 intensity levels go up from around 2.25 to 2.50. It seems that this was quantified at the E3.5. Considering the authors' conclusions regarding the role of Notch signalling prior to the morula stage, it seems that it would be more relevant to investigate the effect of N1ICD in Tead4^-/-^ embryos at earlier stages and it may be that nuclear CDX2 expression is rescued more impressively at these stages. Additionally, to further asses the relative importance of Notch and Tead4 at early stages, the reverse experiment would be to try and rescue nuclear CDX2 expression in Rbpj^-/-^ embryos using a constitutively active form of TEAD4 for instance.

The reviewer rightly points out that there is only a weak effect of N1ICD in rescuing CDX2 expression in *Tead4*^-/-^ embryos. As a matter of fact, and following the advice of reviewer #1, we carefully examined the cell number of the embryos analysed in this experiments. After removing a too early *Tead4*^-/-^ embryo, and despite including two other embryos with a comparable cell number (40-65), the reanalysis showed no significant difference in CDX2 expression (see above response to reviewer #1 and Author response image 1). The suggestion made by the reviewer that changes at the morula stage could be more prominent is certainly of interest. Unfortunately, the few mice from this colony we had available did not produce further litters. This is a complex breeding, which involves three different alleles (*Tead4-, Rosa26-stop-N1ICD-ires-eGFP*, and *Sox2Cre*) that we would need to generate from scratch. The time, space in the mouse facility, and effort this would mean made this experiment not feasible in a reasonable time frame in order to revise the manuscript. Therefore, and in light of the results described above and in the response to reviewer #1, we have decided to remove this experiment from the revised version of the manuscript, We believe this does not affect significantly our work or change the conclusions reached.

As for the reviewer’s suggestion to perform the reverse experiment, that is to rescue the expression of CDX2 in *Rbpj*^-/-^ embryos by forced expression of an active form of TEAD4, this would certainly be of great interest. However, this is an extremely complex experiment from a logistics point of view. We would need to generate and maintain a colony of *Rbpj*^+/-^ mice big enough to generate KO embryos in sufficient number. Due to the lethality of homozygotes, this would require separate heterozygous male (to use as studs) and female (to use for superovulation and the production of embryos) colonies. On top of that, embryos resulting from the het cross would need to be individually microinjected with the TEAD4 construct, immunostained, and imaged. Only after this process could embryos be genotyped, and only one quarter of them would be homozygous KOs. Taking into account the variability and mosaicism inherent to microinjection experiments, the number of mutant embryos that we would need to analyse in order to obtain a robust result would be large. We simply do not have the capacity, in terms of mouse space, funding and time, to carry out this experiment.

5) Lack of clarity in the GOF and LOF regarding recombination eventsIn the GOF and LOF experiments, the authors resort to immunostaining to identify the different cell recombination events. However, the GOF cells were triple positive because of antibody cross-reactivity. In addition, the images presented are unclear and, the examples given, fail to alleviate concerns about the validity of the approach and why it was used. If fixation reduced the fluorescence signal, could the embryos have been imaged live prior to that? Alternatively, have the authors attempted to validate their method by other means (by genotyping after immunostaining of individual dissociated blastomeres for example)? The fact that the less likely recombination event was observed the most often using this method adds to the concern that it may not accurately identify the right type of cell.

We agree with the reviewer that ideally using endogenous fluorescence single to follow the recombination events would have been best. Unfortunately, in this mouse line signal is too weak to observe directly, either in live embryos or fixed specimens. That is why we used immunostaining to distinguish different recombined cells. As for the reviewer’s suggestion to genotype individual cells after immunostaining, although possible in principle, in our opinion it would result extremely challenging. First, dissociation of fixed cells from the embryos (as required for immunostaining) is difficult if not impossible without cross-contamination of neighbouring cells. And second, purification of DNA from this material would again be not that easy.

Nevertheless, the concern raised by the reviewer is valid, so we devised a way to independently test the panel of antibodies used to identify each different recombination event. For this, we used ES cells carrying the same construct, that were separated by FACS according to their fluorescence (fluorescence in cultured cells is bright enough to allow sorting). This is the same strategy used in the experiments described in Figure 6. We then tested all three antibodies (anti-CHERRY, -GFP and –HA) in each of the three sorted populations, finding the expected cross-reactivities: anti-CHERRY recognized the wildtype and gain-of-function recombined cassettes, anti-GFP the loss-of-function and the gain-of-function cassette, and anti-HA the gain-of-function cassette. We have included these results in the revised version in Figure 4—figure supplement 1A. As for the concern of the reviewer that the less likely recombination event was observed the most often, we have addressed this issue by including a new control experiment using an iChrControl-Mosaic line (see response to reviewer #1 above, and Figure 4—figure supplement 3A-C).

6) Number of independent biological replicates in RNA-seq experimentsOnly 2 Rbpj^-/-^ morulae were used in RNA-seq experiments. It seems unreasonable to justify this by the decision of wanting to strictly compare litter mates.

We have increased the sample size of *Rbpj*^-/-^ embryos analysed and repeated the analysis, reaching essentially the same conclusions as before (see response to reviewer #1 above). Updated results are shown in Figure 5, Figure 5—figure supplement 1 and Figure 5—source data 1 and 2, as well as in the GEO database.

7) Quantification of transient transgenic experimentFigure 5E-F – It is not clear why the authors quantified positive and negative embryos. Unless I misunderstood the experiment, it seems that embryos injected with the mutated transcriptional enhancer should either drive H2B-mRFP weakly or not at all in comparison to the wild type construct. As a result, it seems that signal intensity should have been quantified instead.

We deeply apologize for the lack of clarity in the way we explain this set of experiments in the text, as the reviewer rightly points out. Reading again the original version, we can see how we took for granted certain methodological aspect that are far from clear and cannot be readily followed in the description we provide. We will try doing so in the following paragraph, as well as correcting the manuscript in order to provide a better explanation of the experiment.

In these transient transgenic experiments, each embryo is individually injected with the construct at the one-cell stage, allowed to develop to the desired stage (in this case, up to morula) and then scored for activity of the reporter. Because of the technique itself, embryos will be mosaic for the transgene, and we do not have control of the amount of construct received by an individual embryo, or even by different cells of the same embryo. Therefore, we cannot use signal intensity or number of positive cells as a reliable readout of construct activity. We thus score as positive any embryo that shows activity even in a single cell. When we use an empty vector (just the backbone of the minimal promoter plus the reporter), we obtain up to 10-15% of positive embryos. However, when testing fragments showing activity (such as the *Oct4* distal enhancer, or the *Cdx2* TEE), this proportion increases to 2-80%. Examples of this approach can be found in previous publications from our group (Pernaute et al. 2010, Dev Dyn 239 6209; Fernandez-Tresguerres et al. 2010, PNAS 107 19950-60; Rayon et al., 2014; Rayon et al. 2016, Sci Rep 6 27139). In the case of the *Tle4*-up genomic fragment, the wildtype construct drives expression in approximately 30% of microinjected embryos, and thus we consider it as a positive enhancer. However, when we mutate the putative RBPJ binding site only 13% embryos showed reporter activity. We therefore concluded that mutation of this sequence resulted in diminished enhancer activity. (Figure 5E, F) On the other hand, the *Tbx3*-i7 fragment drove reporter activity in 56% of embryos, what did not change significantly upon mutation of the putative RBPJ site (Figure 5—figure supplement 2B; C). In this case, we conclude that this genomic fragment does contain enhancer activity, but this is not dependant exclusively on the RBPJ site.

8) CRISPR experimentFigure 5G – Not enough information is provided to understand what mutations were obtained in the CRISPR experiment. Was a repair construct lacking the enhancer sequence introduced to generate a knockin in the endogenous locus? If not, were the different mutated embryos harbouring the same mutations?

We apologize once more for the lack of experimental detail provided in the original version of the manuscript. For these experiments, we used two gRNA flanking the putative RBPJ binding site, with the aim of generating deletion of approximately 150 bp in both cases. Due to the fact that each individual embryos needs to be processed for genotyping by PCR and expression analysis by RT-qPCR, we cannot obtain enough material to check in the same embryos the exact deletion generated. In order to provide more information on this point, we have carried out new deletion experiments, and in this case used the embryos for genomic DNA extraction, followed by PCR and cloning of the fragments of the expected size for the deletions. Sanger sequencing of individual clones shows that in the case of the deletion of the *Tle4*-up element, we recover two different deletions, and in the case of the *Tbx3*-i7 element three, all within the expected range. This data is now shown in Figure 5—figure supplement 2E, F, and we have corrected the text where necessary.

Reviewer #3:[…] The manuscript would benefit from some clarification regarding the role of Notch signalling in the exit of naive pluripotency. Specifically, it would be important to understand if Rbpj/Notch signalling is required for this process or if the regulation of Prdm14 and Dppa3 is reflecting other roles of Notch signalling. Therefore, it would be interesting to have a broader picture of the role of Notch in exit of pluripotency and if Tle4 is an important target during this process. Using naïve pluripotent ES cells (2i+Lif) and their differentiation will most likely be the most straightforward way to address this question.

This is a very interesting suggestion by the reviewer, so we repeated this experiment with ES cells grown in 2i+LIF. We observe the same trend as when using LIF+serum, in that Notch gain-of-function cells upregulated its putative targets *Tle4* and *Tbx3* following the same trend as established early epiblast markers (*Fgf5, Pou3f1* or *Otx2*). Interestingly, when we examined the expression of later differentiation markers such as *Eomes* or *Gata6*, we did not see any differences related to varying Notch activity. This suggests that Notch signalling is regulating early transition from naïve pluripotency, but not directly specific lineage differentiation events. We have now included these results in the text and present them in Figure 6—figure supplement 4.